# Emergency Decision-Making for Middle Route of South-to-North Water Diversion Project Using Case-Based Reasoning and Prospect Theory

**Feng Li [1], Pengchao Zhang [1], Xin Huang [2,*], Jiabin Sun [1] and Qian Li [1,3]**

[1] School of Water Conservancy, North China University of Water Resources and Electric Power, Zhengzhou 450046, China

[2] School of Agricultural Science and Engineering, Hohai University, Nanjing 211100, China

[3] Research Center for Economy of Upper Reaches of the Yangtze River, Chongqing Technology and Business University, Chongqing 400067, China

* Correspondence: 220210060003@hhu.edu.cn

**Abstract:** The middle route of the South-to-North Water Diversion Project is one of the crucial frameworks of China's water network and an essential channel for water resource allocation in North China. The safe operation of the project has a huge impact on regional economic development, social stability and other aspects. The objectives of this research are to improve the disposal efficiency of all kinds of accidents during the operation of the Middle Route of the South-to-North Water Diversion Project, reduce people's property losses and ensure the safety of water supply along the line. This paper will put forward a new emergency decision-making method based on case-based reasoning technology and prospect theory. The method is divided into two parts: (1) Collecting the historical case information and building the case library. The frame representation in the case-based reasoning technology is used to describe the characteristics of historical cases and adopt the two-level method of historical cases fast retrieval and similarity fuzzy matching retrieval to complete the preliminary selection of emergency plans; (2) The decision-making and optimization model of disposal plans based on prospect theory, namely, using the value function and probability weight classification to measure the prospect value of similar schemes and selecting the optimal disposal scheme, in order to improve the science and rationality of the decision-making results. Finally, examples are taken to verify the feasibility and effectiveness of the method.

**Keywords:** emergency decision-making; south-to-north water diversion project; case-based reasoning; prospect theory

## 1. Introduction

The middle route of the South-to-North Water Diversion Project starts in Nanyang, Henan, through the Huai River watershed of the Yangtze River, and travels north along the western part of the Huang Huaihai Plain to Tuan Cheng Lake in Beijing and the Outer Ring River in Tianjin, shown in Figure 1. It is more than 1400 km long, transferring 9.5 billion cubic meters of water annually, connecting the Yangtze River, the Yellow River, the Huai River and the Hai He River four big basin. This project, which has the main water resources deployment channels for Beijing, Tianjin and even important cities in North China, has improved the current situation of water shortages in North China and promoted the economy, population, society, resources and environment of North and South China. The collaborative development of the country's economy is of great significance to the high-quality development of China's society and economy. However, the middle route of the South-to-North Water transfer project is a long-distance linear water transfer project with complex terrain, landform and geological conditions along the way. It passes through farmland, woodland, villages and cities, and arranges node projects such as crossing canals,

roads and railways through buildings and ensuring the water supply, such as control gates and diversion gates. The management and protection conditions along the route are complex, and there are many risk factors affecting the safety of the water supply. Once it is affected by flood, earthquake, pollution and other emergencies, engineering damage and environmental damage will occur, and water transmission will be interrupted, which will lead to significant economic losses and even casualties. Based on this, the middle line of the South-to-North Water Diversion Project Management Bureau has compiled emergency plans for earthquakes and floods. However, because the time, place and type of emergencies have varying degrees of unpredictability, suddenness and irregularity, the traditional emergency plans cannot cover everything and this limits the timeliness of the actual disposal plan. Therefore, an effective emergency management system must be established to deal with emergencies and avoid or reduce the significant impact of emergencies on the project.

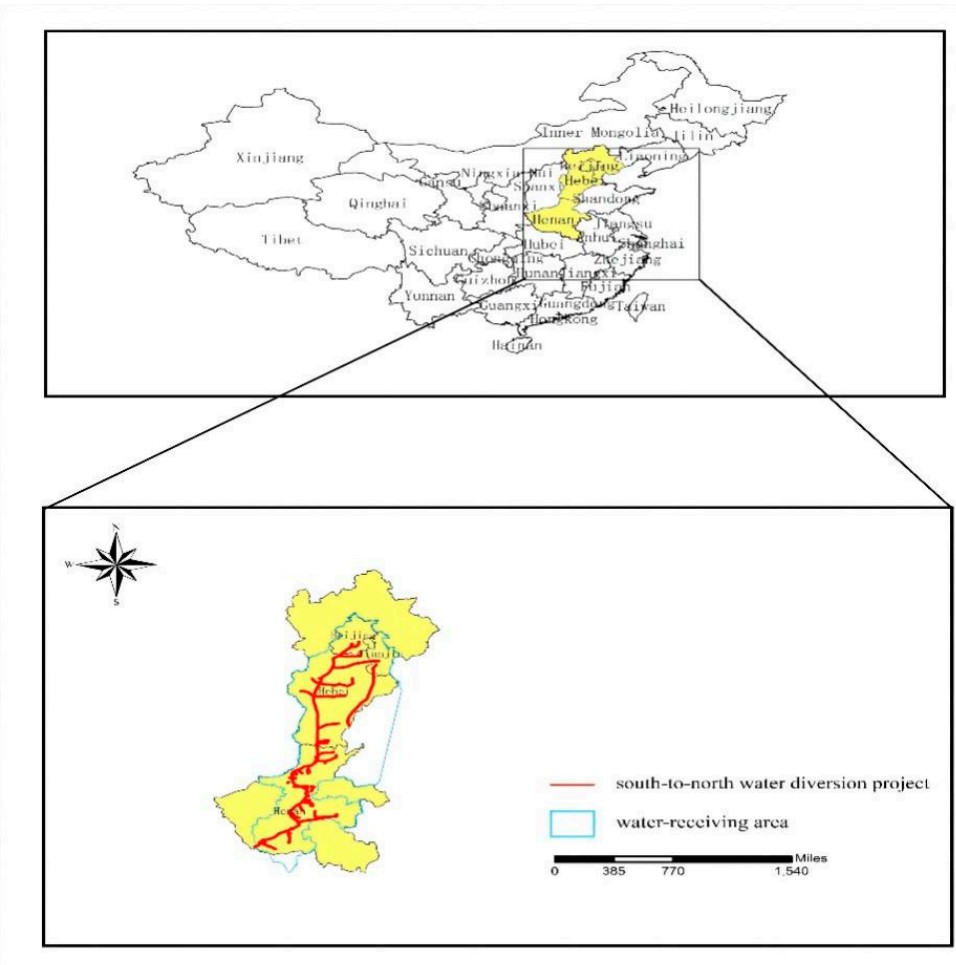

**Figure 1.** Schematic diagram of the middle route of the south-to-north water diversion project.

Based on the analysis of the operating conditions of the middle route of the South-to-North Water Diversion Project, this paper summarizes the engineering safety accident cases and related cases of similar projects and constructs the optimization and selection model for the operation safety accident of the middle route of the South-to-North Water Diversion Project using case reasoning technology and prospect theory. As shown in Figure 2, the model is mainly divided into two parts. The first one is divided into a primary emergency scheme primary based on case reasoning. It includes two parts: case representation and case retrieval. Among them, the case representation is implemented by the framework representation method, which can clarify the content, attribute characteristics and data

characteristics of the case representation, and improve the quality of case retrieval. On this basis, a reasonable search strategy and a screening process are designed to build the emergency scheme primary model. The second part is preferred for an emergency plan. On the basis of the primary election of the emergency plan, considering the certain differences between the historical cases and the current accidents, the disposal plan also has limitations. The site decision-makers need to revise the disposal plan of the historical cases according to the accident situation and then make preferred decisions. Prospect theory, as a method that can effectively evaluate the impact of decision makers' subjective preferences on accident decisions, is for objective factors such as uncertainty in the development of emergencies, incomplete decision-making information, as well as subjective income preferences and loss avoidance of decision-making experts; the psychology of the accident is highly adapted to the situation in which the accident disposal decision is made. Therefore, this paper adopts the prospect theory model to make the optimal decision for the accident disposal plan.

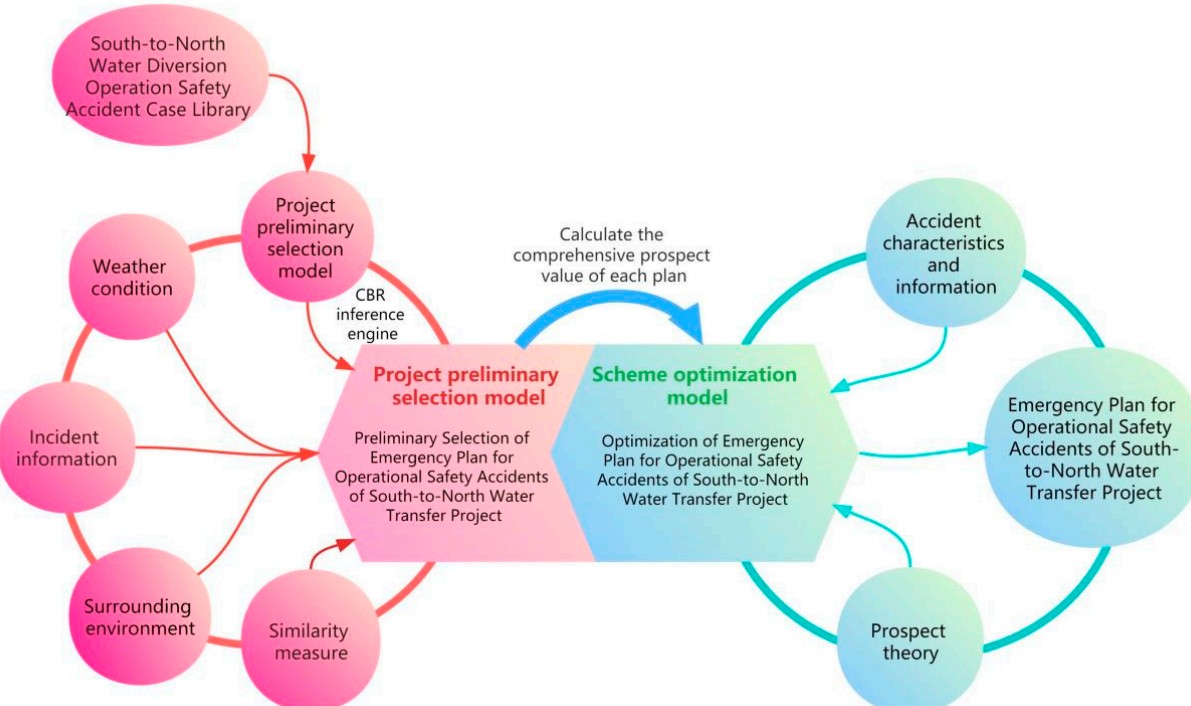

**Figure 2.** Schematic diagram of the selection model of emergency plan for operation safety accident in the middle route of the south-to-north water diversion project.

## 2. Literature Review

### 2.1. Emergency Management Decision Method

Emergency management is an important part of China's national governance system and governance capabilities [1]. Natural disasters and sudden accidents in the engineering industry have brought a series of casualties and economic losses to the country and have had a serious social impact. Therefore, it is of great significance to establish the necessary mechanisms to deal with emergencies through various ways to prevent, deal with and manage emergencies for various industries in today's society. The decision-making around accidents has always been a weak link of industry management. Many scholars have had a lot of research results in the prevention, management and decision-making around accidents. Wu proposed a modified gray correlation analysis method based on the concept of vector entrainment, using gray correlation analysis to deal with complex and variable environments. He optimizes the assessment accuracy by measuring the similarity between the reference and alternative sequences, and proposes an improved consensus-promoting gray correlation analysis method for multi-criteria group decision-making by introducing

expected utility theory as a way to reduce the risk of large-scale catastrophic oil spill accidents [2]. Rong, Liu and Pei based an innovative decision-making algorithm on the generalized image fuzzy Archimedes copula priority operator and a new scoring function, considering that the priority relationship and correlation of the identified attributes are proposed to evaluate the problem of the emergency management scheme [3]. Sun, Mi, Chen and Liu introduced a maximum consensus block and multi-grain decision theory coarse set model to solve the problem of classification accuracy of a simple crude set model and confirm the feasibility of the method with cases of contingency plan selection [4]. Ju, Choi, Choi, Park and Lee designed an emergency decision support system on several modules in the planning, disaster perception and response phase to prepare for a disaster in water-processing infrastructure [5]. Shan, Liu, Wei, Xu, Zhang and Yu designed a multi-stage dynamic evaluation model, and they propose a user-generated content system as a fitting source to support emergency management [6].

### 2.2. Application of Case Reasoning in Emergency Management Decisions

Case-based reasoning is a method of solving problems based on past knowledge. This method compares the current situation with the previous historical cases in the case base to find similar historical cases, and then obtains the solution method of the new problem through the solution method of this similar historical case. It is currently widely used in various fields, including medical treatment [7,8], education [9,10], artificial intelligence [11,12], railway [13,14], engineering project risk management [15,16], aquaculture [17,18] and so on. Case-based reasoning also provides a new direction in solving emergency management decision-making problems and a new perspective for solving emergency management problems. And it has good versatility. This provides conditions for achieving a perfect emergency plan library and rapid emergency response [19]. In recent years, many scholars have had relevant studies in this area: Jiang proposed a decision-making method for building safety risk management based on the case inference method, and improved the reasoning process by integrating the similarity algorithm and the correlation algorithm, and finally, verified it by the safety risk assessment case of subway construction [20]. Gohym described the application of a case-based method in building safety risk identification [19]. Stramr developed a sensitivity analysis method combined with optimized case-based reasoning to improve comparative models between cases [21].

### 2.3. Application of Prospect Theory

Kahneman and Tversky in 1979 proposed prospect theory based on factors such as game theory, psychology and irrational mental changes of decision-makers in the decision-making process [22]. Prospect theory means that before the decision-maker's behavior, the decision-maker will set a reference point in advance according to a plan. When the benefit or loss is near the reference point, the decision-maker prefers to avoid the risk and face losses. In situations, decision-makers are more inclined to take risks; for decision makers, under the same numerical value, the pain caused by loss is far greater than the satisfaction brought by gains. After decades of development, Chinese scholars have been increasingly studying prospect theory. Peng et. al proposed a multi-attribute decision-making method based on prospect theory for stochastic multiple-attribute decision-making problems with trapezoidal fuzzy probabilities and unknown weights [23]. Meng, et. al. used the method of the multi-attribute decision by combining intuitive fuzzy information of the Atanassov interval [24]. Jia proposed a new decision method based on the rough number and prospect theory to solve the multi-criterion group decision problem of risk and uncertainty [25]. Jiang WenQi proposed a multi-criteria decision-making method based on prospect theory and Vikor for risky multi-criteria decision-making problems with fuzzy criteria values [26]. Tong-Tong Nie proposed a discrete stochastic multi-attribute decision method based on cumulative foreground theory and generalized Shapley function is proposed for discrete random multi-attribute decision problems with interval neutral attribute values and incomplete known attribute weights [27]. Wei Xu et al. considered

the finiteness of decision-makers in consensus-building rationality, and proposed a direct consensus framework based on cumulative prospect theory to solve the group decision-making problem with eight preference representation structures [28].

To sum up, in the field of emergency management, on the one hand, most of the research is directed at the emergency management and disposal of a single emergency, without considering the overall perspective; on the other hand, there are few previous studies of decision procedures that consider the psychological factors of decision-makers. Therefore, according to the actual needs of the middle line of the South-to-North Water Diversion Project, this article considers applying the case-reasoning method in the emergency management process, classifying the environmental scenarios of case occurrence and integrating them into a complete case library. When the cases occur, and are integrated into a complete case library, when a new event occurs, one uses case-based reasoning technology to select similar historical cases, and at the same time, considers the psychological factors of decision-makers' loss-aversion, income preference, etc.; and we use the prospect theory ideas in behavioral economics to propose a case-based reasoning and prospect theory emergency decision model.

### 3. Research Process

This article uses a systematic process to illustrate the application process of the proposed method. We divide the method into three major pieces of content:

(1) Case representation. We use a specific method to represent all accident case scenarios, accident case disposal plans and accident case disposal effects as data. The case scenarios mainly include accident characteristic information, meteorological conditions, accident point information and surrounding environmental conditions; disposal plans include information such as rescue equipment, rescue teams, rescue materials, and specific measures; accident case disposal effects include accident disposal results, on-site recovery status and lessons learned;

(2) Case-based reasoning scheme selection model. First, the historical accident cases are input into the case database, and the first-level retrieval is carried out. Then, the historical case set is mainly screened according to the characteristic attributes of high discrimination; the overall similarity between the selected cases and the target case is compared. It consists of three parts: attributes similarity, structural similarity and attributes weight. Finally, several case sets similar to the target accident are obtained;

(3) Scheme optimization model based on prospect theory. For the candidate case set selected by the primary selection model, standardize the data, set the reference point and the probability of the situation state, determine the probability weight function value, and finally, calculate the comprehensive prospect value of each scheme, and sort them to obtain the optimal scheme. The research process is shown in Figure 3.

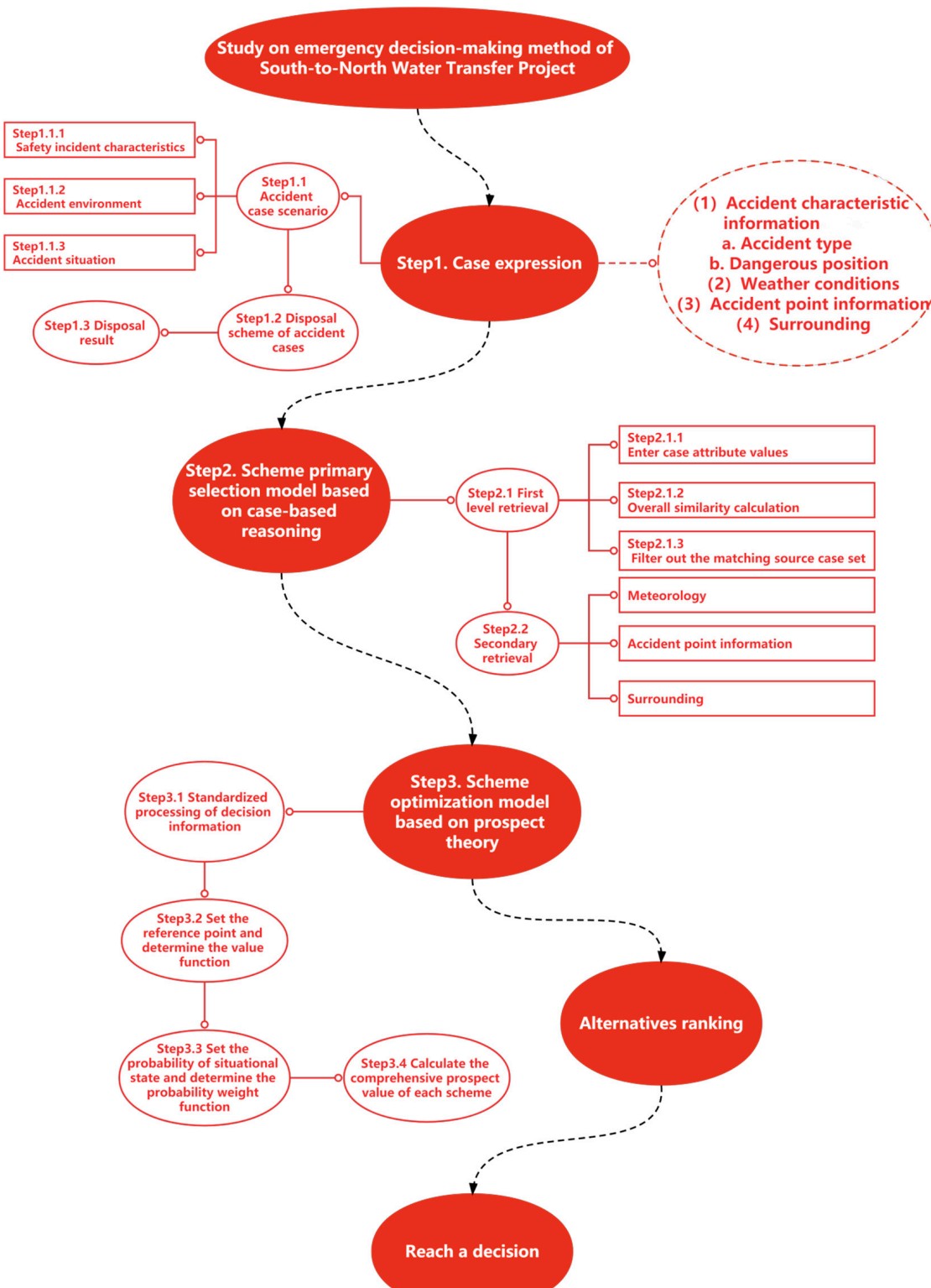

**Figure 3.** Schematic diagram of the research process.

## 4. Research Method

### 4.1. Case Representation

4.1.1. The Basic Content of the Case

Summarizing the cases in the case database is the prerequisite for emergency decision-making. The purpose of a case presentation is to have a clear and detailed record of the accident. A complete case consists of three parts:

1.  Accident case scenario. Recording the characteristics of the operation safety accident, the occurrence environment, the accident situation and other information of the accident completely, to fully reflect the accident situation state at the scene.
2.  Accident case disposal plan. Recording the detailed and feasible disposal plan contents of the specific accidents, including the rescue equipment, rescue team, rescue materials and specific disposal measures.
3.  Disposal effect of accident cases. After the completion of the on-site rescue work, the accident disposal effect will be summarized and evaluated, including the final results of the disposal plan, the recovery of the damaged site, lessons and experience of lessons, etc.

The core of the case-based reasoning technology is to calculate the similarity between the current event and the historical event based on the situation of the accident case, filter out the historical cases with high similarity, then adjust and apply the treatment plan of similar cases according to the current situation of the accident. At the same time, supplement the disposal effect after the implementation of the plan, and finally, complete the disposal of the entire accident.

4.1.2. Characteristic Description of Operational Safety Accidents

According to the characteristics of engineering accidents during the middle route of the South-to-North Water Diversion Project, the characteristic attributes describing the status of such accidents will be divided into four parts:

1.  Accident characteristic information. Contains two types of attributes: accident type and dangerous location.
2.  Meteorologic condition. Includes weather, temperature, wind direction, wind force and visibility, etc.
3.  Accident point information. Includes the occurrence time, casualties, damage degree, the length of the affected canal section, water transmission operation status and other specific information.
4.  Surrounding environmental conditions. Include the characteristics of canal sections, emergency road conditions, distance from residential areas, the number of residents nearby, secondary disasters, the presence of public buildings and other information.

4.1.3. Framework Representation of Operational Safety Accident Cases

Case representation is mainly divided into framework representation [29], XML representation [30], ontology representation [31], etc. Because the framework representation has a strong structure and clear logic, this article applies the framework representation to express engineering accidents. The case representation can be divided into three parts:

1.  Description of the case (accident) scenario. It mainly includes accident characteristics information, meteorological conditions, accident point information and surrounding environmental conditions. As shown in Table 1.
2.  Description of the case disposal plan. Includes rescue equipment, rescue teams, rescue supplies, specific measures and other information. As shown in Table 2.
3.  Description of the case disposal effect: including accident disposal results, on-site recovery and experience, etc. As shown in Table 3.

**Table 1.** Operational safety accident case scenario representation model.

| Frame Name | Slots (First Level Attribute) | Side (Second Level Attribute) | Side Value Description | Side Value Type |
|---|---|---|---|---|
| Feature Represen-tation of Accident Scenario | Slot 1: Accident characteristic information | Side 1: Type of accident [32] | Value: Slope instability, Foundation instability, Leakage damage, etc. | Symbol type |
| | | Side 2: Dangerous location [32] | Value: External slope of channel, Internal slope of canal embankment, Canal embankment filling, etc. | Symbol type |
| | Slot 2: Weather condition | Side 1: Weather [33] | Value: Sunny, Cloudy, Overcast, etc. | Symbol type |
| | | Side 2: Temperature [33] | Value: °C | Interval type |
| | | Side 3: Wind force [33] | Value: Wind level | Enumeration |
| | | Side 4: Visibility [33] | Value: Extremely, Bad, good, etc. | Fuzzy linguistic type |
| | Slot 3: Incident information | Side 1: Time of occurrence [34] | Value: Accident occurrence time | Numeric type |
| | | Side 2: Number of casualties [34] | Value: Number of people | Numeric type |
| | | Side 3: Degree of damage [35] | Value: Mild, More serious, Serious, etc. | Fuzzy linguistic type |
| | | Side 4: Affect the length of the canal section [35] | Value: Specific length | Numeric type |
| | | Side 5: Water delivery operation status [35] | Value: Normal, Abnormal | Symbol type |
| | Slot 4: Surrounding environment | Side 1: Channel characteristics [36] | Value: Excavation canal section, Filled canal section, etc. | Symbol type |
| | | Side 2: Emergency road conditions [36] | Value: Unobstructed, blocked | Symbol type |
| | | Side 3: Distance from the residential area [35] | Value: Length | Numeric type |
| | | Side 4: Number of nearby residents [36] | Value: Number of people | Numeric type |
| | | Side 5: Secondary disaster status [37] | Value: Mild, More serious, Serious, etc. | Fuzzy linguistic type |
| | | Side 6: Whether there are public buildings [37] | Value: yes, no | Symbol type |

**Table 2.** Accident Handling Plan Repr;sentation Model.

| Frame Name | Slot | Slot Value |
|---|---|---|
| Emergency rescue plan said | Slot 1: Rescue equipment | Value: Excavator, Forklift, Compaction machinery, Vibration roller, etc. |
| | Slot 2: Rescue team | Value: Commander, Rescue construction team, Medical rescue team, etc. |
| | Slot 3: Rescue supplies | Value: Woven bags, Geomembrane, Geotextile, etc. |
| | Slot 4: Specific measures | Value: Detailed description of the disposal measures |

**Table 3.** Accident disposal effect presentation model.

| Frame Name | Slot | Slot Value |
|---|---|---|
| Expression of emergency rescue plan disposal effect | Slot 1: Disposal result | Value: Description of the actual effect of the disposal plan |
| | Slot 2: On-site recovery status | Value: Bad, Worse, General, Better, Good |
| | Slot 3: Lessons | Value: Summarizing the accident |

*4.2. Preliminary Selection Model of the Scheme Based on Case-Based Reasoning*

4.2.1. Case Retrieval Strategy and Process

Case Retrieval Strategy

Case retrieval is generally sorted based on the overall similarity between the target cases and historical cases. The overall similarity is usually composed of three parts: attribute similarity, structural similarity and attribute weight. With the increasing number of cases in the system, the scheme screening only with the attribute similarity between cases will lead to the retrieval rate and accuracy. At the same time, due to the complex categories and number of the middle route of the South-to-North Water Diversion Project, this article improves the efficiency of case retrieval by constructing a two-level retrieval strategy. The retrieval process is shown in Figure 4.

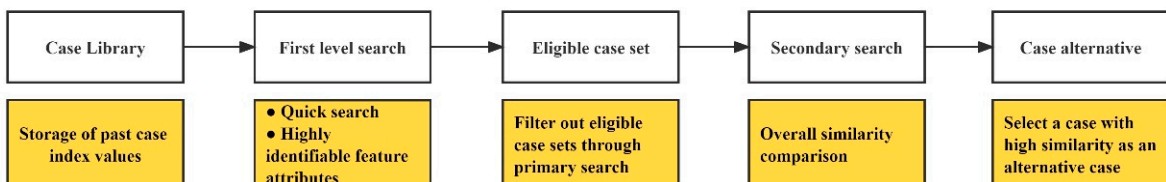

**Figure 4.** Case retrieval process.

1. First level search: Quick search

Select highly recognizable feature attributes as first-level retrieval attributes, and filter out the set of historical cases that meet the conditions. According to the characteristics of operation safety accidents in the middle route of the South-to-North Water Diversion Project, the accident type and dangerous site are selected as the first-level retrieval attribute.

2. Secondary search: Overall similarity comparison

The historical case set obtained through first-level retrieval will be compared with the target case. Then, the historical cases which are similar to the target case will be selected to provide a reference for the disposal of current dangerous accidents. This article selects the sub-attributes of meteorological conditions, accident point information and surrounding environmental conditions as secondary retrieval attributes.

Case Organization Index

Introducing the case index can speed up case retrieval and improve the operation efficiency of case inference. This article adopts an inductive index strategy to construct a hierarchical index structure based on case-key properties. Considering that in the process of the middle route of the South-to-North Water Diversion Project, the formulation of the accident disposal plan has obvious correspondence with the dangerous site and accident type. The dangerous site and accident type are taken as the key attributes of the accident cases of the middle route of the South-to-North Water Diversion Project, as the index structure is shown in Figure 5 below.

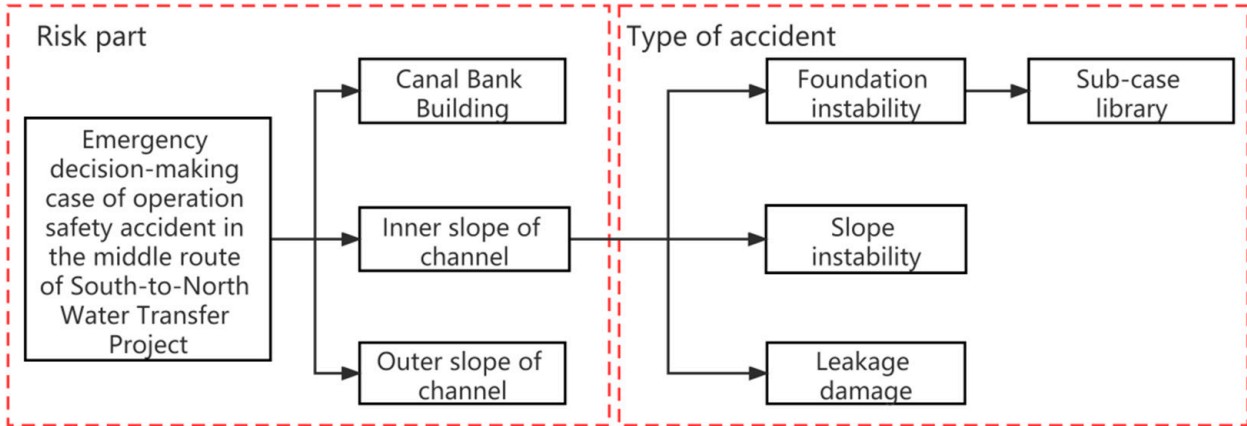

**Figure 5.** Index structure diagram.

Case Selection Process

Extract the attribute value to enter it into the retrieval matching model by obtaining the detailed information of the current accident case, and screen similar cases and correct the disposal scheme by matching the overall similarity. Finally, the most reasonable disposal scheme is preferred through expert decision-making. The emergency decision-making process based on case-based reasoning is shown in Figure 6.

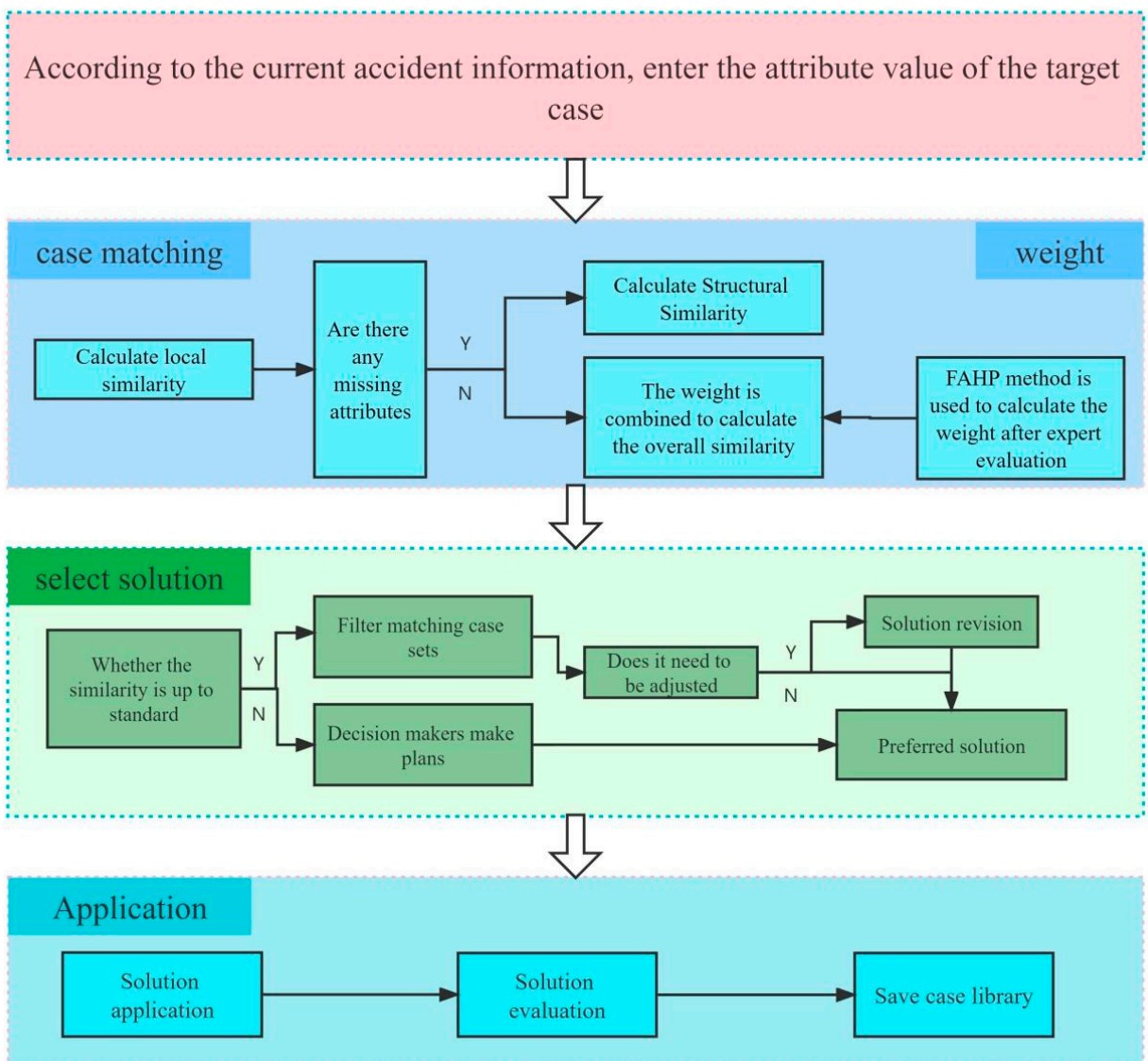

**Figure 6.** Case-based reasoning scheme screening flowchart.

### 4.2.2. Calculation of the Case Attribute Weight

The fuzzy analytic hierarchy is a qualitative and quantitative analysis method. Using the relative importance of the pairwise comparisons between the same level indicators to determine the weights can improve the scientific nature of the weight calculations [38].

1.  Build a hierarchical structure model

This paper constructs a hierarchical structure model of safety accident operation in the middle route of the South-to-North Water Diversion Project as shown in Figure 7.

2.  Construct fuzzy judgment matrix

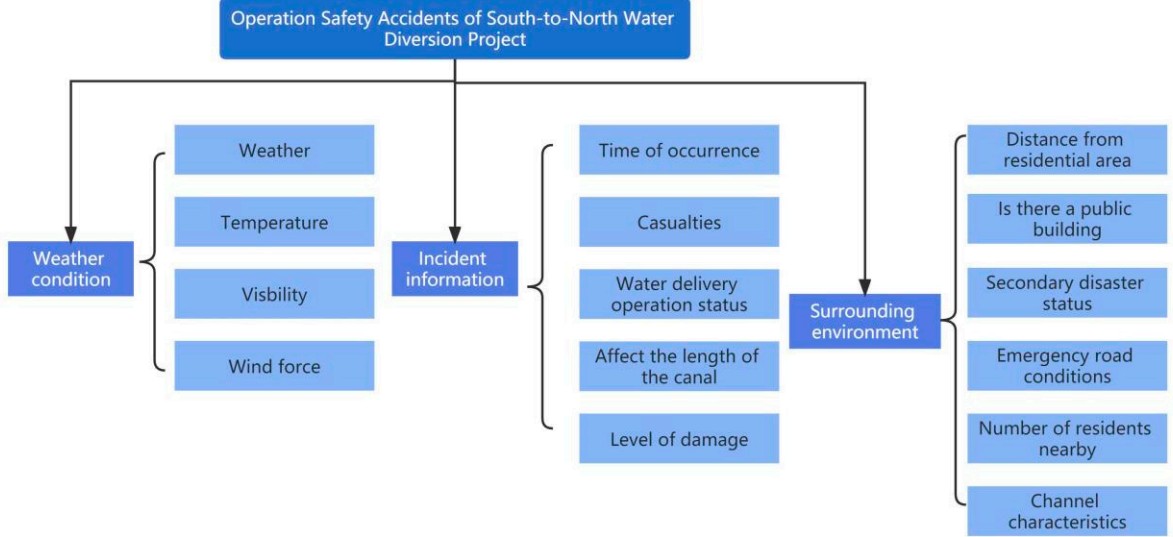

**Figure 7.** Operational Security Incident Hierarchy Model.

Suppose the index set of a certain layer is $A = \{A_1, A_2, \cdots, A_n\}$, then invite experts to score the two index levels respectively, then perform secondary processing according to the fuzzy scale in Table 4; the fuzzy judgment matrix R of index set A can be obtained as:

$$R = \begin{bmatrix} r_{11} & r_{12} & \cdots & r_{1n} \\ r_{21} & r_{22} & \cdots & r_{2n} \\ \vdots & \vdots & \ddots & \vdots \\ r_{n1} & r_{n2} & \cdots & r_{nn} \end{bmatrix} \tag{1}$$

**Table 4.** Fuzzy scale and meaning.

| $r_{ij}$ | The Meaning of $r_{ij}$ | $r_{ij}$ |
|---|---|---|
| 0.5 | Elements $r_i$ and $r_j$ are equally important | 0.5 |
| 0.6 | The element $r_i$ is slightly more important than the element $r_j$ | 0.4 |
| 0.7 | The element $r_i$ is more important than the element $r_j$ | 0.3 |
| 0.8 | Element $r_i$ is stronger than element $r_j$ | 0.2 |
| 0.9 | Element $r_i$ is far more important than element $r_j$ | 0.1 |

In the formula, $r_{ij}$ is expressed as the comparison result of the importance of element $r_i$ and element $r_j$; take the value according to Table 4.

If the fuzzy judgment matrix $R = (r_{ij})_{n \times n}$ satisfies $r_{ij} = r_{ik} - r_{jk} + 0.5$ for any $i,j,k, k = 1, 2, 3, \cdots, n$, it is called a fuzzy consistent judgment matrix. If not, the fuzzy consistent judgment matrix $R'$ is obtained by Equation (2). The fuzzy consistent judgment matrix is used to calculate the weight, which effectively avoids the consistency check caused by the subjectivity of judgment.

$$r'_{ij} = \frac{1}{2(n-1)} \left( \sum_{k=1}^{n} r_{ik} - \sum_{k=1}^{n} r_{jk} \right) + 0.5, \ i, j = 1, 2, \cdots, n \tag{2}$$

In the formula, $r'_{ij}$ is the value in the standardized fuzzy consistent judgment matrix $R'$, and $n$ is the number of attribute indicators.

3. Calculate the weight of each attribute index

Combining the least squares method of finding the sorting vector, the index weight can be obtained as:

$$w_i = \frac{1}{n} - \frac{n}{4\alpha(n-1)} + \frac{1}{2\alpha(n-1)} \sum_{j=1}^{n} r'_{ij}, \; i, j = 1, 2, \cdots, n \tag{3}$$

In the formula, $w_i$ represents the weight of the $i$ index, $r'_{ij}$ represents the value in the fuzzy consensus judgment matrix $R'$, and $n$ is the number of attribute indicators. To ensure the difference between the weight values of the relative importance of the factors, set $\alpha = 2(n-1)/5$ here.

### 4.2.3. Calculation of Overall Similarity between Cases

Due to the complex environment of the accident site in the middle route of the South-to-North Water Diversion Project, some accident attribute values may be missing. The structural similarity is adopted here to solve the problem that the overall similarity calculation method caused by such conditions is not applicable [39].

1.  Structural similarity calculation between cases

First, calculate the sets $A$ and $B$ composed of the non-empty attributes of the target case $X$ and the historical case $Y$, respectively, then calculate the intersection $C = A \cap B$ and the union $D = A \cup B$ of $A$ and $B$, then calculate the sum of the weights $W_1 = \sum_{i \in C} w_i$, $W_2 = \sum_{j \in D} w_j$ of all the attribute weights in $C$ and $D$, and finally, define the structural similarity $S$ between the target case $X$ and the historical case $Y$ as:

$$S = W_1 / W_2 \tag{4}$$

2.  Local attribute similarity calculation

Its attribute value types include numeric, enumerated, symbolic, interval and fuzzy language types.

(1) Numerical type similarity algorithm

The spatial distance between numbers and numbers is often used to represent the similarity of the determined number properties. Inter-property similarity was calculated based on the Haiming distance formula [40]:

$$sim(X_j, Y_{ij}) = 1 - dist(X_j, Y_{ij}) = 1 - |x_j - y_{ij}| / |\max(j) - \min(j)| \tag{5}$$

In the formula, $X_j$ represents the $j$ attribute of the target case $X$; $Y_{ij}$ represents the $j$ attribute of the historical case $Y_i$; $sim(X_j, Y_{ij})$ represents the similarity between the target case $X$ and the historical case $Y_i$ in the $j$ attribute; $x_j$ and $y_{ij}$ are the target case $X$, respectively, data corresponding to the $j$ attribute of historical case $Y_i$; $\min(j)$ and $\max(j)$ represent the minimum and maximum values of attribute $j$ in the case library.

(2) Enumeration-type similarity algorithm

This kind of attribute is valued in a data set with a certain hierarchical relationship, and its similarity calculation formula is:

$$sim(X_j, Y_{ij}) = 1 - \left| \frac{x_j - y_{ij}}{g} \right| \tag{6}$$

In the formula, $sim(X_j, Y_{ij})$ represents the similarity between the target case $X$ and the historical case $Y_i$ in the $j$ attribute; g is the number of levels of the value of the attribute $j$.

(3) Symbolic-type similarity algorithm

Symbol attribute is usually a deterministic symbolic description, and when the attribute value is the same, defining the attribute similarity is 1, not 0.

$$sim(X_j, Y_{ij}) = \begin{cases} 1, x_j = y_{ij} \\ 0, x_j \neq y_{ij} \end{cases} \tag{7}$$

In the formula, $sim(X_j, Y_{ij})$ represents the similarity between the target case $X$ and the historical case $Y_i$ in the $j$ attribute; $x_j$ and $y_{ij}$, respectively, represent the symbol attribute value corresponding to the attribute $j$ of the target case $X$ and the historical case $Y_i$.

(4) Interval number-type similarity algorithm

When the collated data are not the exact value, it is necessary to use interval numbers to describe the specific information of the attributes [34]. The calculation formula can be expressed as:

$$sim(X_j, Y_{ij}) = 1 - \frac{\int_{a_1}^{a_2} \int_{b_1}^{b_2} |y - x| dy dx}{(\max(j) - \min(j))(a_2 - a_1)(b_2 - b_1)} \tag{8}$$

In the formula, $sim(X_j, Y_{ij})$ represents the similarity between the target case $X$ and the historical case $Y_i$ on the $j$ attribute; $[a_1, a_2]$ and $[b_1, b_2]$ are the interval values corresponding to the $j$ attribute of the target case $X_i$ and the historical case $Y_{ij}$, respectively, $\min(j)$ and $\max(j)$, respectively, represent the minimum value of the attribute $j$ and maximum value: $\min(j) \leq a_1, a_2, b_1, b_2 \leq \max(j)$.

(5) Fuzzy language-type similarity algorithm

Fuzzy language variables are used to describe the fuzziness of attribute values [41]. The similarity calculation method of fuzzy linguistic variables is to convert linguistic variables into triangular fuzzy numbers $(a_{ij}, b_{ij}, c_{ij})$, select 1, 3, 5, 7, 9 scale values to convert linguistic variables into triangular fuzzy number evaluation values, then use Equation (9) and Equations (4)–(10), standardizing it as $(x_{ij}, y_{ij}, z_{ij})$.

$$(x_{ij}, y_{ij}, z_{ij}) = \left( \frac{a_{ij}}{\sqrt{\sum\limits_{i=1}^{m} (c_{ij})^2}}, \frac{b_{ij}}{\sqrt{\sum\limits_{i=1}^{m} (c_{ij})^2}}, \frac{c_{ij}}{\sqrt{\sum\limits_{i=1}^{m} (a_{ij})^2}}, a_{ij}, b_{ij}, c_{ij} \in T^{\alpha} \right) \tag{9}$$

$$(x_{ij}, y_{ij}, z_{ij}) = \left( \frac{1}{c_{ij}\sqrt{\sum\limits_{i=1}^{m} (\frac{1}{a_{ij}})^2}}, \frac{1}{b_{ij}\sqrt{\sum\limits_{i=1}^{m} (\frac{1}{b_{ij}})^2}}, \frac{1}{b_{ij}\sqrt{\sum\limits_{i=1}^{m} (\frac{1}{c_{ij}})^2}}, a_{ij}, b_{ij}, c_{ij} \in T^{\beta} \right) \tag{10}$$

In the formula, $T^{\alpha}$ and $T^{\beta}$, respectively, indicate that the attribute value is of benefit type and cost type. The triangular fuzzy number $(a_{ij}, b_{ij}, c_{ij})$ represents the score value of the $j$ attribute of the $i$ case, where $a_{ij} \leq b_{ij} \leq c_{ij}, a_{ij}, b_{ij}, c_{ij}$ represent the lower, median and upper bounds of the triangular fuzzy number, respectively.

The normalized triangular fuzzy number $(x_{ij}, y_{ij}, z_{ij})$ is transformed into an interval number $[s_{ij}^-, s_{ij}^+]$, and the interval similarity calculation method (4-8) is used to obtain the similarity between fuzzy language attributes [42,43]. Among them: $s_{ij}^- = (x_{ij} + y_{ij})/2$, $s_{ij}^+ = (y_{ij} + z_{ij})/2$. The triangular fuzzy number rating corresponding to this language variable is shown in Table 5:

3. Overall similarity calculation

**Table 5.** Triangle fuzzy number ratings corresponding to secondary disaster severity.

| Language Variables | Fuzzy Rating |
|---|---|
| Has/Have no influence | (1, 1, 3) |
| Light | (1, 3, 5) |
| Heavy | (3, 5, 7) |
| Serious | (5, 7, 9) |
| Very serious | (7, 9, 9) |

The overall similarity is based on the first-level retrieval, combined with the local attribute similarity and structure similarity of the secondary properties.

(1) Similarity calculation of first-level retrieval attributes:

The similarity calculation formula for the first-level search is:

$$Sim^1(X, Y) = \prod sim_t(X, Y) \tag{11}$$

In the formula, $t = 1, 2$ respectively represent the accident type and the attributes of the dangerous location.

(2) Similarity calculation of secondary search attributes

There are various secondary retrieval attributes, different attribute value types, and the calculation formula is:

$$Sim^2(X, Y) = \frac{\sum\limits_{i=1}^{n} \sum\limits_{j=1}^{m_i} w_i w_{ij} \times sim(x_{ij}, y_{ij})}{\sum\limits_{i=1}^{n} \sum\limits_{j=1}^{m_i} w_i w_{ij}} \tag{12}$$

In the formula, $Sim^2(X, Y)$ is the similarity value of the second-level retrieval attributes of the target case $X$ and the historical case $Y$; $sim(x_{ij}, y_{ij})$ is the second-level attribute similarity of the first-level attribute index $i$ of each case $X$ and the case $Y$; $w_i$ is the second-level attribute similarity given to the first-level attribute, the weight of $i$; $w_{ij}$ is the weight assigned to the secondary attribute index $j$ under the primary attribute $i$.

(3) Calculate the overall similarity between the target case and each historical case

Due to the lack of attribute values, the structural similarity is introduced to calculate the overall similarity $Sim(X, Y)$ between cases as:

$$Sim(X, Y) = Sim^1(X, Y) \times Sim^2(X, Y) \times S \tag{13}$$

*4.3. Preferred Model Based on Foreground Theory*

Considering the uncertainty in the evolution of the accident scenario, the ambiguity of the decision information and the decision experts, the preferred model of the South-to-North Water Diversion Project is based on the risk preference of the prospect theory. Firstly, the data are standardized; secondly, the positive and negative ideal points are set as reference points to calculate the value function value of each scheme. Then, the set risk probability weight function is used to determine the event evolution state probability. Finally, the value function value and the scenario probability weight are worth ranking for the comprehensive foreground value of each scheme and to select the optimal disposal scheme. The flow chart is shown in Figure 8.

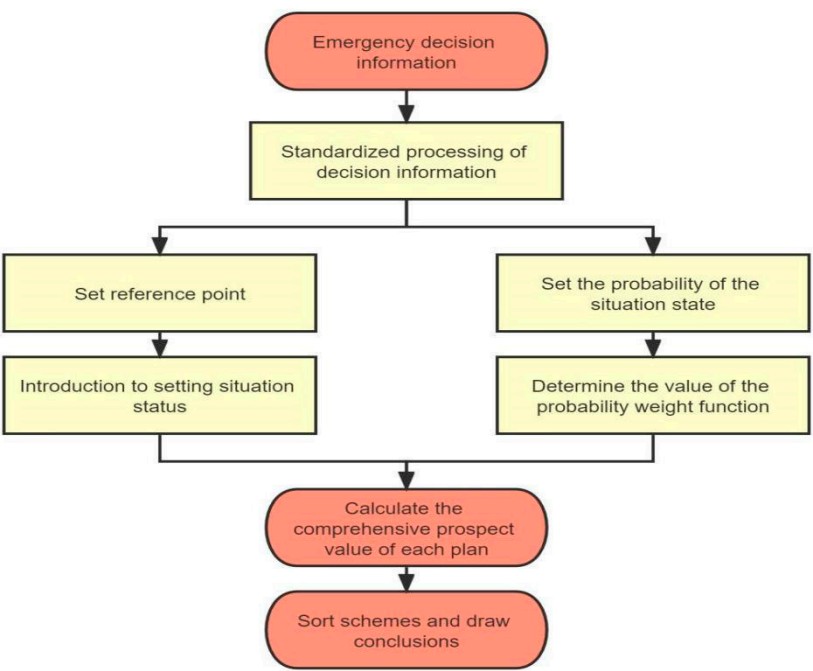

**Figure 8.** A preferred flow chart of the emergency decision plan.

4.3.1. Standardized Processing of Decision-Making Information

Case-based reasoning technology is used to screen out the decision-making plan set $D = (D_1, D_2, \cdots, D_m)$, and the attribute value of the decision-making plan $D_i$ under the emergency decision index $C_j$ and the scenario evolution state $S_k$ is set to $a_{ijk} = [a^l_{ijk}, a^u_{ijk}]$. When decision-makers make optimal decisions on emergency plans, they first adopt the ratio method to standardize the treatment. According to the characteristics of the attribute value, it is divided into benefit attribute value and cost attribute value. The formula is as follows:

(1) The attribute value is the benefit type:

$$r^L_{ijk} = \frac{a^L_{ijk}}{\sqrt{\sum\limits_{i=1}^{m} \left(a^U_{ijk}\right)^2}}, r^U_{ijk} = \frac{a^U_{ijk}}{\sqrt{\sum\limits_{i=1}^{m} \left(a^L_{ijk}\right)^2}} \tag{14}$$

(2) The attribute value is of a cost type:

$$r^L_{ijk} = \frac{1/a^U_{ijk}}{\sqrt{\sum\limits_{i=1}^{m} \left(1/a^L_{ijk}\right)^2}}, r^U_{ijk} = \frac{1/a^L_{ijk}}{\sqrt{\sum\limits_{i=1}^{m} \left(1/a^U_{ijk}\right)^2}} \tag{15}$$

4.3.2. Setting of the Reference Point

Prospect theory emphasizes the gains and losses of the decision results relative to the reference point, rather than the absolute value of the decision results themselves [44]. The setting of the reference point is the basis for the calculation of the value function and affects the decision maker's judgment of gains and losses. In this thesis, the positive and negative ideal points are selected as reference points. Ideal positive and negative point values are shown in Equations (16) and (4–17):

$$r^+_{ijk} = [r^{+L}_{ijk}, r^{+U}_{ijk}] = [\max(r^L_{ijk}), \max(r^U_{ijk})] \tag{16}$$

$$r^-_{ijk} = [r^{-L}_{ijk}, r^{-U}_{ijk}] = [\min(r^L_{ijk}), \min(r^U_{ijk})] \tag{17}$$

The gains and losses in the value function are expressed by calculating the distance between the scheme attribute values and the corresponding positive and negative ideal points. In the situational state $S_k$, the distance between the normalized attribute value of the $j$ attribute of each emergency decision plan $D_i$ and the positive and negative ideal points is $d(r_{ijk}, r_{ijk}^+)$ and $d(r_{ijk}, r_{ijk}^-)$, respectively.

$$d(r_{ijk}, r_{ijk}^+) = \sqrt{(r_{ijk}^L - r_{ijk}^{+L})^2 + (r_{ijk}^U - r_{ijk}^{+U})^2} \tag{18}$$

$$d(r_{ijk}, r_{ijk}^-) = \sqrt{(r_{ijk}^L - r_{ijk}^{-L})^2 + (r_{ijk}^U - r_{ijk}^{-U})^2} \tag{19}$$

4.3.3. Calculate the Value of the Value Function

In the prospect theory, the value function $v(\triangle x_i)$ is used to replace the utility function in the expected utility theory, shown in Figure 9. The prospect theory considers the evaluation of the gain or loss of a reference point, which is the value of the subjective perception of the decision maker.

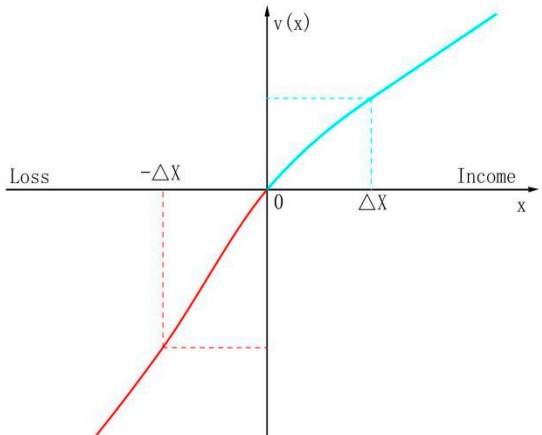

**Figure 9.** Value function.

The value function is defined as follows:

$$v(\triangle x_i) = \begin{cases} (\triangle x_i)^\alpha & \triangle x_i \geq 0 \\ -\lambda(-\triangle x_i)^\beta & \triangle x_i < 0 \end{cases} \tag{20}$$

In the formula, $\triangle x_i = x_i - x_0$ and $\triangle x_i$ are the profit and loss value of the decision-making plan attribute $x_i$ compared to the reference point $x_0$, and $\triangle x_i \geq 0$ means income, $\triangle x_i < 0$ means loss; $\alpha$, $\beta$ are the risk preference coefficient of decision-makers, $0 < \alpha, \beta < 1$, $\alpha$, $\beta$. greater, represents the decision-maker more prone to risk, $\lambda$ is the loss-avoidance factor, and the larger the value of $\lambda(\lambda > 1)$, the more averse the decision-maker is to risk loss. Kahneman and Tversky give parameter values through a large number of experimental results, and set $\alpha = \beta = 0.88$ and $\lambda = 2.25$.

By analyzing the actual meaning of the value function and combining the positive and negative ideal point distance in Equations (18) and (19), the value function of the $j$ attribute of the emergency decision plan $D_i$ under the scenario state $S_k$ is calculated:

$$v^+(d(r_{ijk}, r_{ijk}^-)) = d(r_{ijk}, r_{ijk}^-)^\beta \tag{21}$$

$$v^-(d(r_{ijk}, r_{ijk}^+)) = -\theta d(r_{ijk}, r_{ijk}^+)^\alpha \tag{22}$$

In the formula, $\alpha$ and $\beta(\alpha \geq 0, \beta \leq 1)$ represent the degree of preference when the value function gains or loses, and $\theta$ represents the risk aversion coefficient.

### 4.3.4. Calculate Probability Weights

The probability weight function in foreground theory is a psychological perception of the decision-maker on the probability of a certain accident state. According to the one-site conditions and personal experience, the accident disposal experts give the situation $S = (S_1, S_2, \cdots, S_t)$ that the accident may evolve. The probability of each situation is $P = (P_1, P_2, \cdots, P_t)$. The probability weighting functions of the gains and losses are set to $w^+(P_k)$ and $w^-(P_k)$, respectively. The calculation rules for the probability weighting function are as follows:

$$w^+(P_k) = \frac{P_k^{\gamma}}{\left(P_k^{\gamma} + (1 - P_k)^{\gamma}\right)^{1/\gamma}} \tag{23}$$

$$w^-(P_k) = \frac{P_k^{\delta}}{\left(P_k^{\delta} + (1 - P_k)^{\delta}\right)^{1/\delta}} \tag{24}$$

In the formula, $w^+(P_k)$ and $w^-(P_k)$ are the non-linear probability weights for gains and losses, respectively. $P_k$ is the probability of event occurrence, $\gamma$ is the coefficient of risk-return attitude and $\delta$ is the coefficient of the risk-loss attitude, respectively, and the values of $0 < \gamma$, $\delta < 1$, $\gamma$ and $\delta$ are set to $\gamma = 0.61$, $\delta = 0.69$.

### 4.3.5. Identify the Optimal Scheme

According to the prospect theory, the prospect value of the $j$ attribute of scheme $D_i(i = 1, 2, \cdots, n)$ can be obtained as:

$$V_{ij} = \sum_{k=1}^{t} v^+(d(r_{ijk}, r_{ijk}^-))w^+(P_k) + \sum_{k=1}^{t} v^-(d(r_{ijk}, r_{ijk}^+))w^-(P_k) \tag{25}$$

Finally, combine the weight $w_j(j = 1, 2, \cdots, n)$ of the indicator to calculate the comprehensive prospect value $V_i$ of the plan $D_i$:

$$V_i = \sum_{j=1}^{n} w_j V_{ij} \tag{26}$$

According to the comprehensive prospect value $V_i$, the emergency decision-making plans are sorted. The larger the value of $V_i$, the better the plan and the more satisfied the decision-maker.

## 5. Case Analysis

### 5.1. Preliminary Selection of Emergency Plan

#### 5.1.1. Accident Description

By collecting and sorting out the documents, the case data of the emergency management decision-making event of the water transfer project in recent years have been taken as the historical case, and the accident case database for emergency management decision-making was established according to the characteristics and attributes of the South-to-North Water Diversion Project and the corresponding index system. The channel landslide event of a management office of the South-to-North Water Diversion Project was taken as an example to verify the case retrieval algorithm and scheme generation method of this article. Accident overview: The accident canal section is located 550 m upstream of a management office of the South-to-North Water Diversion Project., and the hydrogeological situation of the left bank is complex. The preliminary design and survey results show that the lithology of the canal slope is mainly composed of loess-heavy silty loam, silty clay, pebble and clay rock. The design flow rate of this section is 260 m$^3$/s, with

an increased flow of 310 m$^3$/s. The design longitudinal slope is 1/20000~1/28000, the water crossing section is trapezoidal, the bottom of the channel is 14.5 m wide and the top of the embankment is 5 m wide. The inner slope below the first-grade horse lane is lined with a single slope, which has a slope pitch of 1:2.0, cast-in-place concrete, lining thickness of 8 cm, lining concrete strength grade of C20, frost resistance label F150 and impermeability label W6. The channel lining sub-seam spacing is 4m, the through seam and half seam spacing are arranged, and the seam width is 2 cm. A total of 600 g/m$^2$ composite is laid under the lining plate, with a membrane thickness of 0.3 mm. The inner slope above the first-class horse road shall be protected by a precast concrete hexagonal hollow frame and be equipped with a concrete shoulder guard, a frame filled with soil and grass planting. On 24 June 2016, heavy rainfall occurred from 2 a.m. to 11 a.m., with temperatures falling to 16 to 25 °C, causing poor visibility. At about 16:25 on 25 June, the management personnel found that the inner slope of the left bank slope of the canal section had collapsed locally, with a damaged area of 55 m$^2$. The affected canal section is about 20 m, and the left bank of the first horse road near the curb side of the asphalt road subsidence, the asphalt road surface and the top of the original replacement body partially emptied, with no casualties. Remember the current case as $X_0$, and take 10 existing event cases from the historical case database to form a historical case set $Y = (Y_1, Y_2, Y_3, Y_4, Y_5, Y_6, Y_7, Y_8, Y_9, Y_{10})$. The attribute values of the historical case and the current case are shown in Table 6.

**Table 6.** Target case and historical case attribute values.

| Attribute Metrics | CaseY$_1$ | CaseY$_2$ | CaseY$_3$ | CaseY$_4$ | CaseY$_5$ | CaseY$_6$ | CaseY$_7$ | CaseY$_8$ | CaseY$_9$ | CaseY$_{10}$ | CaseX$_0$ |
|---|---|---|---|---|---|---|---|---|---|---|---|
| Type of accident | Landslide | Crack | Landslide | Landslide | Destroy by rush of water | Leakage | Landslide | Landslide | Landslide | Landslide | Landslide |
| Dangerous parts | Channel inner slope | Lining board | Channel inner slope | Channel inner slope | Flood protection embankment | Aqueduct | Channel inner slope | Channel inner slope | Channel outer slope | Channel inner slope | Channel inner slope |
| Weather | Heavy rain | Koyuki | Heavy rain | Heavy fog | Rainstorm | Fine | / | Fine | Heavy rain | Light rain | Heavy rain |
| Temperature | 28–35 °C | −3–11 °C | 14–21 °C | 24–32 °C | 21–29 °C | 5–12 °C | 17–26 °C | 18–22 °C | 20–26 °C | 6–12 °C | 16–25 °C |
| Wind-force | 8 | 1 | 4 | / | 4 | / | 3 | 1 | 4 | / | 6 |
| Visibility | / | Difference | Very bad | Difference | Difference | Good | Better | / | Difference | Generally | Difference |
| Occurrence time | 19:45 | 10:31 | 11:39 | 3:52 | 3:18 | 9:48 | 7:35 | 13:26 | 16:50 | 17:10 | 16:25 |
| Casualty number | 3 | 2 | 3 | 0 | 5 | 0 | / | 0 | 5 | 4 | 2 |
| Damaged condition | Extremely serious | Heavy | Slight | More serious | Serious | Generally | / | Extremely serious | Generally | Serious | Heavy |
| Affect the length of the canal | 130 m | 0 m | 50 m | 30 m | 80 m | 0 m | 60 m | / | / | 50 m | 20 m |
| Water transmission operation condition | Normal | Normal | Normal | Normal | Abnormal | Normal | Abnormal | Normal | Normal | / | Normal |
| Channel characteristics | Dig square canal section | / | Dig square canal section | Dig square canal section | Dig Square canal section | / | Dig square canal section | Dig square canal section | Fill canal section | Dig square canal section | Dig square canal section |
| Emergency road condition | Abnormal | Normal | Abnormal | Normal | Abnormal | Normal | Abnormal | Normal | Normal | Abnormal | Normal |
| Distance from residential areas | 900 m | 1300 m | 500 m | 1000 m | 3000 m | 1800 m | 3100 m | 800 m | 2000 m | 2300 m | 600 m |
| Number of residents | 452 | 368 | 1364 | 2358 | 682 | 1561 | 3562 | 2355 | 1985 | 1465 | / |
| Secondary disaster | None | None | None | Slight | Serious | None | Heavier | Heavier | None | Serious | Generally |
| Are there public buildings | Have | None | None | None | None | None | Have | None | None | Have | None |

Taking the attribute index "visibility" as an example; it is divided by fuzzy language variables, referring to the classification standard and actual needs of Horizontal Visibility, and is divided into five standards: extremely poor, poor, relatively poor, general and excellent according to the visual distance. "Visibility ≤ 50 m" is set to "extremely poor", "50 m ≤ Visibility ≤ 500 m" set to "poor", "500 m ≤ visibility ≤ 1000 m" set to "relatively poor", "1000 m ≤ visibility ≤ 2000 m" set to "general", "Visibility ≥ 2000 m" set to "good".

Because the "occurrence time" of the accident is described by the 24 h system, it is necessary to quantify the attribute value of the "occurrence time" to improve the scientific nature of the attribute similarity calculation, where "8 a.m.~12 a.m., 2 p.m.~6 p.m." is defined as "1", "12 a.m.~14 p.m." is defined as "2", "6 a.m.~9 a.m., 6 p.m.~8 p.m." is quantified as "3" and "8 p.m.~0 a.m., 0 a.m.~6 a.m." is quantified as "4".

Using the inductive index method in the case library can speed up the case retrieval and improve the operation efficiency of case inference. Because the dangerous site and accident type are the key indicators of the South-to-North Water emergency management decision, the case library and the target case of accident type and case are consistent as a sub-case library, according to the target case of accident type and danger site, using the inductive index technology in the case database accident type for landslide, danger as channel slope accident cases $Y_1$, cases $Y_3$, cases $Y_4$, cases $Y_7$, cases $Y_8$, cases $Y_{10}$ and constructed into a sub-case library.

### 5.1.2. Determination of the Index Weights

(1) First-level index weight

According to the importance of each index, the emergency decision-makers of the South-to-North Water Diversion Project compared the first-level attribute indexes, namely, meteorological conditions, accident point information and the surrounding environmental conditions, then obtained a fuzzy judgment matrix of

$$R = \begin{bmatrix} 0.5 & 0.2 & 0.3 \\ 0.8 & 0.5 & 0.6 \\ 0.7 & 0.4 & 0.5 \end{bmatrix}$$

The weight values of each index calculated by Equations (2) and (3) are shown in Table 7.

**Table 7.** First-level index weights.

| First Level Index | Meteorologic Condition | Accident Point Information | The Surrounding Environment |
|---|---|---|---|
| weight | 0.216 | 0.427 | 0.357 |

(2) Secondary index weight

The determination of secondary index weights is consistent with the calculation of primary index weights. Combined with the weight results of the first-level index, the weights of the second-level index values under the meteorological conditions, the accident point information and the surrounding environmental conditions are calculated by Equations (2) and (3) are shown in Table 8.

**Table 8.** Secondary index weights.

| Secondary Index | Weight | Secondary Index | Weight | Secondary Index | Weight |
|---|---|---|---|---|---|
| Weather | 0.064 | Casualty number | 0.102 | Emergency road condition | 0.067 |
| Temperature | 0.050 | Damaged condition | 0.088 | Distance from residential areas | 0.061 |
| Wind-force | 0.048 | Influence channel length | 0.075 | The number of nearby residents | 0.061 |
| Visibility | 0.054 | Water transmission operation condition | 0.094 | Secondary disaster situation | 0.063 |
| Occurrence time | 0.069 | Channel characteristics | 0.054 | Are there public buildings | 0.052 |

### 5.1.3. Similarity Calculation between Cases

When conducting case retrieval using the similarity calculation method of this article, the similarity threshold size is first set to select the number of N similar cases that meet the requirements according to the threshold size, and set the similarity threshold > 0.6. According to the above local similarity calculation method (5)~(10), the target case has no empty local similarity with each historical case, and the calculation results are shown in Table 9.

Due to the incomplete attribute value in accident cases, the attribute index of the missing attribute value cannot participate in the similarity calculation. At the same time, calculating the similarity between the attribute values alone cannot accurately describe the similarity between the historical cases and the target cases in the case library. Therefore, structural similarity is introduced to make up for the unscientific overall similarity calculation caused by the lack of the attribute value. Take the similarity calculation of historical cases $Y_1$ and target cases $X_0$ as an example; historical case $Y_1$ has missing visibility attribute values, while the target case $X_0$ has missing attribute values for the number of nearby residents. According to the calculation Equation (4), the structural similarity of the two cases is 0.887, while the similarity between the two cases is shown in Table 9, and the overall

similarity is 0.559 according to the global similarity calculation Equation (13). Similarly, the results of the structural similarity and global similarity calculation between the target cases and other historical cases are obtained, as shown in Table 10.

**Table 9.** Local attribute similarity between cases.

| Secondary Attributes | $X_0$ and $Y_1$ | $X_0$ and $Y_3$ | $X_0$ and $Y_4$ | $X_0$ and $Y_7$ | $X_0$ and $Y_8$ | $X_0$ and $Y_{10}$ |
|---|---|---|---|---|---|---|
| Weather | 1 | 0 | 1 | Null | 0 | 0 |
| Temperature | 0.621 | 0.874 | 0.741 | 0.893 | 0.916 | 0.603 |
| Wind-force | 0.833 | 0.833 | Null | 0.75 | 0.583 | Null |
| Visibility | Null | 0.83 | 1 | 0.578 | Null | 0.779 |
| Occurrence time | 0.5 | 1 | 0 | 0.5 | 0.75 | 1 |
| Casualty number | 0.75 | 0.50 | 0.75 | Null | 0.50 | 0.50 |
| damaged condition | 0.760 | 0.634 | 1 | Null | 0.760 | 0.831 |
| Affect the length of the canal | 0.154 | 0.923 | 0.923 | 0.692 | Null | 0.769 |
| Water transmission operation condition | 1 | 1 | 1 | 0 | 1 | Null |
| Channel characteristics | 1 | 1 | 1 | 1 | 1 | 1 |
| Emergency road condition | 0 | 1 | 1 | 0 | 1 | 0 |
| Distance from residential areas | 0.885 | 0.962 | 0.846 | 0.039 | 0.923 | 0.346 |
| Secondary disaster | 0.530 | Null | 1 | 0.890 | 0.530 | 0.860 |
| Are there public buildings | 0 | 1 | 1 | 0 | 1 | 0 |

**Table 10.** Structure and overall similarity between cases.

| Case | $X_0$ and $Y_1$ | $X_0$ and $Y_3$ | $X_0$ and $Y_4$ | $X_0$ and $Y_7$ | $X_0$ and $Y_8$ | $X_0$ and $Y_{10}$ |
|---|---|---|---|---|---|---|
| Structural similarity | 0.887 | 0.9411 | 0.8931 | 0.6871 | 0.8121 | 0.7991 |
| Overall similarity | 0.559 | 0.7436 | 0.6809 | 0.3108 | 0.6228 | 0.4524 |

As can be seen from Table 10, the overall similarity of historical cases $Y_3$, historical cases $Y_4$, historical cases $Y_8$ and target cases $X_0$ is 0.7436, 0.6809, 0.6228, respectively, ranking among the top three in the calculation of similarity between cases, meeting the requirement of a threshold greater than 0.6 in case screening. Therefore, the emergency management decision-making methods of cases $Y_3$, $Y_4$ and $Y_8$ can be used as a reference for target case $X_0$. Case $Y_1$, Case $Y_7$ and Case $Y_{10}$ have low similarities, and their emergency management decision-making methods are not considered.

Through the analysis of the case scenario, the structural similarity of historical cases $Y_1$ is relatively high. However, due to its low attribute similarity in affecting the length of sections, emergency road conditions, secondary disasters and the absence of public buildings, the overall similarity is reduced, and the length of the affected section, emergency road conditions and secondary disasters are important reference factors for formulating emergency rescue plans and mobilizing emergency rescue teams, which also determines that the reference ability of the accident case is extremely low; with historical case $Y_7$, it is because the structural similarity and attribute similarity are both low, which leads to the low overall similarity of the case and the limited reference value of the disposal plan. The historical case $Y_{10}$ is related to the number of casualties, water delivery conditions, emergency road conditions and residential areas. The similarity of distance and the presence or absence of public buildings to the target case $X_0$ is low. The number of casualties is often used to determine the level of accidents. The operation status of the water conveyance is used to judge the impact on the water transfer project. The similarity of these key elements is low. The availability of its corresponding disposal plan is also limited. Historical cases $Y_3$, $Y_4$ and $Y_8$ have high similarity in terms of structural similarity and attribute similarity, and their corresponding disposal plans have a high reference value for formulating current rescue plans.

In addition, by comparing and analyzing the attribute values of historical cases $Y_3, Y_4$ and $Y_8$, it is found that the reason for the high similarity of historical case $Y_3$ is that its information content is relatively sufficient, and the attribute value information under weather conditions is similar to that of the target case $X_0$. The situation is relatively close, while the high similarity of historical case $Y_4$ is due to its high attribute similarity in the accident site information and surrounding environmental conditions; while the high similarity of historical case $Y_8$ is due to its various attributes. It is relatively close to the target case, so the calculation result of the overall similarity also meets the threshold requirement. From the above analysis, it can be seen that due to the impact of weather conditions, accident point information, surrounding environmental conditions, etc., the accident handling scenarios of each plan are different, and their corresponding handling plans are also different. It is also necessary to combine the site conditions and the risk attitudes, bounded rationality and other conditions of decision-making experts to make judgments, so that the plan is more reasonable and the handling of accidents is more effective.

*5.2. Emergency Plan Is Preferred*

5.2.1. Description of the Emergency Plan and Attribute Indicators

After using case-based reasoning technology to select similar historical case sets $Y_3$, $Y_4$ and $Y_8$, the emergency command team also needs to adjust the historical case set $Y_3$, $Y_4$ and $Y_8$ based on the characteristics of the accident and surrounding environmental conditions. Then, they get the new disposal decision plan $D_1$, $D_2$ and $D_3$, the specific content of which is as follows:

$D_1$: For the lining board, adopt the measure of crushing by a gravel bag to temporarily raise the operating water level of the channel to balance the external water pressure. After the treatment of replacing the viscous soil layer, the road surface of the first-grade road is restored. The road structure is asphalt concrete pavement with an emulsified asphalt sealing layer.

$D_2$: Arrange that blind drainage ditches are arranged on the first-level horse road to reduce the groundwater head outside the canal; remove part of the replacement soil set clay interception tooth grooves, and set anchor rods on the slope of the first-level horse road to improve the anti-sliding stability of the replacement soil.

$D_3$: Clean up the soil layer within the scope of the damage and disturbance, fill it with foam concrete, and restore the original section with ordinary concrete as the channel lining panel and the first-grade horse-lane road foundation.

Due to the bad weather and complex geological conditions at the time of the accident, the location of the accident is relatively remote; heavy rain will lead to a muddy and slippery road, and it is difficult for rescue materials and rescued personnel to get in and out. These factors will affect the speed of rescue. In order to ensure the completion of the rescue work as soon as possible, combine the characteristics of the project accident itself and the site construction conditions; it is estimated that the accident may appear with the following three conditions:

$S_1$: The weather conditions get better, the rescue channel will be opened in time to ensure the timely entry of emergency rescue materials and equipment, the geological conditions of the accident site are relatively stable, there will be no continuous landslide phenomenon, and the accident will be timely and effective controlled.

$S_2$: Weather conditions may change, rescue channels may be basically smooth, with emergency rescue supplies and rescue personnel entering the field in succession; but affected by the rain, the geological condition of the construction site is not conducive to rapid rescue. Rescue personnel report on the spot that the accident section channel slope soil is loose and there is the possibility of a secondary landslide. In order to prevent the deterioration of the accident, it is necessary to accelerate the rescue.

$S_3$: The weather continues to deteriorate, the rescue channel is obstructed, all kinds of rescue supplies and rescue personnel cannot enter the site in time, and the slope of the

channel is loose. Affected by heavy rainfall, a secondary landslide, the rescue situation is not ideal, and the difficulty of rescue increases, seriously affecting the channel water safety.

The emergency command team combines the accident site information and expert discussion to infer the probabilities of the above three states as $P_1 = 0.3$, $P_2 = 0.45$, $P_2 = 0.25$.

According to the disposal objectives of the operation safety accidents in the middle route of the South-to-North Water Diversion Project and the characteristics of the disposal decision, the emergency command team decided to evaluate the preferred emergency decision plan from the following aspects, and to set the decision-making criteria as follows in $C_1$, $C_2$, $C_3$ and $C_4$.

$C_1$: Represents the economic cost. In order to complete the emergency resources consumed by the emergency rescue work, the decision-makers prefer to spend the smaller emergency rescue plan with the same emergency rescue effect.

$C_2$: Represents the disaster site control. When dealing with the operation safety accidents of the middle route of the South-to-North Water Diversion Project, the primary task is to control the accident site. Due to the different disposal methods adopted in the emergency decision-making plan, the ability to control the site and curb the development of the accident is also different.

$C_3$: Represents the timeliness of rescue. The timeliness of emergency rescue mainly considers the time required to implement the plan. The speed of implementation of the emergency rescue plan will have an impact on the effect of the emergency rescue work. The score of rescue timeliness is divided into five grades: "poor, relatively poor, general, good and great". The specific scores are obtained from the numerical transformation of fuzzy language variables.

$C_4$: Represents the social impact, mainly referring to the impact of the implementation of the emergency rescue programs on the normal water transmission. The emergency decision-making experts are more inclined to the emergency decision-making plans that can minimize the social impact. The social impact score is divided into four grades: "none, slight, relatively serious and serious", and the specific scores are also obtained from the transformation of fuzzy language variable values.

5.2.2. Calculation of the Prospect Value of Each Scheme

In the initial stage of the operation safety accident, according to the collected accident case information, emergency decision-making experts can retrieve similar case reasoning technology in the historical case library, then take all disposal schemes in the historical case set as reference schemes, and adjust the disposal plan accordingly to form a case set, in order to provide conditions for the preferred decision of the accident disposal scheme. Because the decision information is dynamically changing, it is difficult to give specific values for the evaluation value of each specific scheme. This article adopts the evaluation value of each scheme under various states and indicators. The interval numbers of the emergency decision matrices in Tables 11 and 12.

**Table 11.** Emergency decision matrix of decision criteria $C_1$ and $C_2$.

|  | $C_1$ | | | $C_2$ | | |
|  | $S_1$ | $S_2$ | $S_3$ | $S_1$ | $S_2$ | $S_3$ |
|---|---|---|---|---|---|---|
| $D_1$ | (35, 42) | (41, 50) | (48, 55) | (0.73, 0.81) | (0.65, 0.76) | (0.58, 0.66) |
| $D_2$ | (25, 36) | (32, 41) | (39, 45) | (0.65, 0.71) | (0.58, 0.66) | (0.51, 0.59) |
| $D_3$ | (40, 52) | (53, 61) | (56, 65) | (0.85, 0.93) | (0.78, 0.86) | (0.65, 0.71) |

**Table 12.** Emergency decision matrix of decision criteria $C_3$ and $C_4$.

| | $C_3$ | | | $C_4$ | | |
| | $S_1$ | $S_2$ | $S_3$ | $S_1$ | $S_2$ | $S_3$ |
|---|---|---|---|---|---|---|
| $D_1$ | (0.68, 0.75) | (0.61, 0.67) | (0.55, 0.63) | (0.36, 0.51) | (0.48, 0.55) | (0.53, 0.59) |
| $D_2$ | (0.73, 0.81) | (0.69, 0.75) | (0.61, 0.68) | (0.69, 0.76) | (0.75, 0.82) | (0.79, 0.88) |
| $D_3$ | (0.62, 0.69) | (0.58, 0.63) | (0.51, 0.57) | (0.53, 0.62) | (0.59, 0.65) | (0.62, 0.71) |

Combining the emergency solution preferred model to solve the solution preferred problem in the emergency decision process, the specific calculation procedure is shown as follows:

The attribute values are normalized according to Equations (14) and (15), with the processing results shown below in Tables 13 and 14.

**Table 13.** Standardized decision matrix of decision criteria $C_1$ and $C_2$.

| | $C_1$ | | | $C_2$ | | |
| | $S_1$ | $S_2$ | $S_3$ | $S_1$ | $S_2$ | $S_3$ |
|---|---|---|---|---|---|---|
| $D_1$ | (0.43, 0.69) | (0.46, 0.69) | (0.48, 0.64) | (0.51, 0.63) | (0.49, 0.65) | (0.51, 0.65) |
| $D_2$ | (0.50, 0.97) | (0.56, 0.88) | (0.59, 0.79) | (0.46, 0.55) | (0.44, 0.56) | (0.45, 0.58) |
| $D_3$ | (0.35, 0.60) | (0.37, 0.53) | (0.41, 0.55) | (0.60, 0.72) | (0.59, 0.74) | (0.57, 0.70) |

**Table 14.** Standardized decision matrix of decision criteria $C_3$ and $C_4$.

| | $C_3$ | | | $C_4$ | | |
| | $S_1$ | $S_2$ | $S_3$ | $S_1$ | $S_2$ | $S_3$ |
|---|---|---|---|---|---|---|
| $D_1$ | (0.52, 0.63) | (0.54, 0.64) | (0.51, 0.65) | (0.54, 0.97) | (0.60, 0.78) | (0.61, 0.76) |
| $D_2$ | (0.48, 0.59) | (0.48, 0.57) | (0.47, 0.59) | (0.36, 0.51) | (0.41, 0.50) | (0.41, 0.51) |
| $D_3$ | (0.56, 0.69) | (0.57, 0.67) | (0.56, 0.70) | (0.44, 0.66) | (0.63, 0.0.51) | (0.51, 0.65) |

Then, the positive and negative ideal points are set according to Equations (16) and (17) and the distance between each decision scheme, and negative ideal points are calculated using Equations (18) and (19). The results are shown in Tables 15 and 16.

**Table 15.** Decision criteria, positive and negative ideal distances $C_1$ and $C_2$.

| | $C_1$ | | | | | | $C_2$ | | | | | |
| | $S_1$ | | $S_2$ | | $S_3$ | | $S_1$ | | $S_2$ | | $S_3$ | |
| | $d^+$ | $d^-$ | $d^+$ | $d^-$ | $d^+$ | $d^-$ | $d^+$ | $d^-$ | $d^+$ | $d^-$ | $d^+$ | $d^-$ |
|---|---|---|---|---|---|---|---|---|---|---|---|---|
| $D_1$ | 0.32 | 0.18 | 0.31 | 0.19 | 0.35 | 0.17 | 0.14 | 0.11 | 0.14 | 0.11 | 0.12 | 0.13 |
| $D_2$ | 0.09 | 0.46 | 0.10 | 0.41 | 0.18 | 0.35 | 0.23 | 0.02 | 0.23 | 0.02 | 0.21 | 0.04 |
| $D_3$ | 0.44 | 0.07 | 0.49 | 0.02 | 0.46 | 0.06 | 0.02 | 0.23 | 0.01 | 0.24 | 0.04 | 0.21 |

**Table 16.** Decision criteria, positive and negative ideal distances $C_3$ and $C_4$.

| | $C_3$ | | | | | | $C_4$ | | | | | |
| | $S_1$ | | $S_2$ | | $S_3$ | | $S_1$ | | $S_2$ | | $S_3$ | |
| | $d^+$ | $d^-$ | $d^+$ | $d^-$ | $d^+$ | $d^-$ | $d^+$ | $d^-$ | $d^+$ | $d^-$ | $d^+$ | $d^-$ |
|---|---|---|---|---|---|---|---|---|---|---|---|---|
| $D_1$ | 0.09 | 0.08 | 0.07 | 0.1 | 0.08 | 0.09 | 0.07 | 0.5 | 0.19 | 0.37 | 0.21 | 0.36 |
| $D_2$ | 0.15 | 0.02 | 0.16 | 0.01 | 0.15 | 0.02 | 0.53 | 0.01 | 0.51 | 0.05 | 0.50 | 0.05 |
| $D_3$ | 0.01 | 0.16 | 0.03 | 0.15 | 0.01 | 0.16 | 0.35 | 0.18 | 0.35 | 0.20 | 0.34 | 0.21 |

Next, the positive and negative values of each emergency decision scheme are calculated by using Equations (21) and (22). The specific results are shown in Tables 17 and 18.

**Table 17.** Positive and negative values of decision criteria $C_1$ and $C_2$.

| | $C_1$ | | | | | | $C_2$ | | | | | |
| | $S_1$ | | $S_2$ | | $S_3$ | | $S_1$ | | $S_2$ | | $S_3$ | |
| | $v^+$ | $v^-$ | $v^+$ | $v^-$ | $v^+$ | $v^-$ | $v^+$ | $v^-$ | $v^+$ | $v^-$ | $v^+$ | $v^-$ |
|---|---|---|---|---|---|---|---|---|---|---|---|---|
| $D_1$ | 0.11 | −0.79 | 0.13 | −0.77 | 0.12 | −0.85 | 0.55 | −0.37 | 0.42 | −0.36 | 0.41 | −0.32 |
| $D_2$ | 0.04 | −0.24 | 0.02 | −0.26 | 0.03 | −0.47 | 0.02 | −0.59 | 0.07 | −0.59 | 0.07 | −0.54 |
| $D_3$ | 0.20 | −1.1 | 0.19 | −0.12 | 0.20 | −1.1 | 0.22 | −0.05 | 0.25 | −0.03 | 0.25 | −0.12 |

**Table 18.** Positive and negative values of decision criteria $C_3$ and $C_4$.

| | $C_3$ | | | | | | $C_4$ | | | | | |
| | $S_1$ | | $S_2$ | | $S_3$ | | $S_1$ | | $S_2$ | | $S_3$ | |
| | $v^+$ | $v^-$ | $v^+$ | $v^-$ | $v^+$ | $v^-$ | $v^+$ | $v^-$ | $v^+$ | $v^-$ | $v^+$ | $v^-$ |
|---|---|---|---|---|---|---|---|---|---|---|---|---|
| $D_1$ | 0.11 | −0.24 | 0.13 | −0.20 | 0.12 | −0.22 | 0.55 | −0.2 | 0.42 | −0.49 | 0.41 | −0.54 |
| $D_2$ | 0.04 | −0.38 | 0.02 | −0.43 | 0.03 | −0.4 | 0.02 | −1.25 | 0.07 | −1.22 | 0.07 | −1.19 |
| $D_3$ | 0.20 | −0.04 | 0.19 | −0.09 | 0.20 | −0.03 | 0.22 | −0.86 | 0.25 | −0.86 | 0.25 | −0.83 |

The probability weights of each evolution scenario are calculated using Equations (23) and (24), The results are shown in Table 19.

**Table 19.** Probabilistic weight values for state loss and gain in each scenario.

| | $S_1$ | $S_2$ | $S_3$ |
|---|---|---|---|
| $w^+$ | 0.32 | 0.40 | 0.29 |
| $w^-$ | 0.33 | 0.42 | 0.29 |

Then, the foreground values of each emergency decision plan are calculated combined with Equation (25) and the specific values are shown in the following Table 20:

**Table 20.** Prospect and value table of emergency decision plan $D_1$, $D_2$, $D_3$.

| . | $C_1$ | $C_2$ | $C_3$ | $C_4$ |
|---|---|---|---|---|
| $D_1$ | −0.71 | 0.09 | −0.11 | 0.03 |
| $D_2$ | −0.30 | −0.55 | −0.40 | −1.2 |
| $D_3$ | −0.96 | 0.18 | 0.14 | −0.65 |

The fuzzy analytic hierarchy process is used to calculate the weight of each criterion:

According to the importance of each index, the emergency decision-making management experts compared four indicators: economic cost, disaster site control degree, rescue timeliness and social impact, and obtained the fuzzy judgment matrix as

$$R = \begin{bmatrix} 0.5 & 0.3 & 0.4 & 0.6 \\ 0.7 & 0.5 & 0.6 & 0.7 \\ 0.6 & 0.4 & 0.5 & 0.7 \\ 0.4 & 0.3 & 0.3 & 0.5 \end{bmatrix}$$

The weight of each decision attribute index is obtained as shown in the following Table 21:

**Table 21.** Decision attribute index weights.

| Decision Indicators | Economic Costs | Disaster-Site Control Degree | Rescue Timeliness | Social Influence |
|---|---|---|---|---|
| Weight | 0.23 | 0.30 | 0.27 | 0.20 |

Finally, the comprehensive prospect value is calculated based on Equation (26):

$$D_1 = -0.1579;$$
$$D_2 = -0.5844;$$
$$D_3 = -0.2606;$$

It can be seen that the comprehensive prospect value of each emergency decision plan is sorted as $D_1 > D_3 > D_2$, so the scheme should be selected as $D_1$.

## 6. Comparative Analysis

In order to verify the differences between prospect theory and traditional objective methods, in this section, we select two decision methods: the TOPSIS method and fuzzy comprehensive evaluation method. We apply them to the scheme optimization stage based on the completion of the case primary, and compare the results with the prospect theory in this article.

In the analysis process of the above two objective decision-making methods, first we need to preprocess the interval numbers of the emergency decision matrices to transform them to fixed values. Here, we adopt the method of taking the median value. The second step is to standardize them and then substitute them into the formulas of the two methods.

The TOPSIS (Technique for the order preference by similarity to an ideal solution) method, also known as the "approximating ideal solution sorting method", is based on the relative closeness of the evaluation object and the ideal solution, then ranks the evaluation objects as a whole to determine the relative advantages and disadvantages. It has the advantages of a flexible and convenient calculation process and accurate and reasonable evaluation results [45]. Using this method, in the first step, we use Equation (27) to take the positive and negative ideal solutions $V_j^*$ and $V_j^-$ for the standardized emergency decision matrix; in the second step, use Equation (28) to calculate the indicators corresponding to plan $D_i$, to the distance between the ideal points $S_i^*$ and $S_i^-$; the third step is to calculate the relative closeness $C_i^*$ between the scheme $D_i$ and the positive and negative ideal solutions according to Equation (29), where $0 \leq C_i^* \leq 1$, the positive ideal point $C_i^* = 1$, negative ideal point $C_i^* = 0$; the higher the relative closeness, the better the solution.

$$\begin{cases} V_j^* = \max\{v_{1j}, v_{2j}, \cdots v_{mj}\} \\ V_j^- = \min\{v_{1j}, v_{2j}, \cdots v_{mj}\} \end{cases} \tag{27}$$

$$\begin{cases} S_i^* = \sqrt{\sum\limits_{j=1}^{n} (v_{ij} - v_j^*)^2} \\ S_i^- = \sqrt{\sum\limits_{j=1}^{n} (v_{ij} - v_j^-)^2} \end{cases} \tag{28}$$

$$C_i^* = \frac{S_i^-}{S_i^- + S_i^+} \tag{29}$$

The fuzzy comprehensive evaluation method is a comprehensive evaluation method and decision-making method based on fuzzy mathematics. It describes the boundary fuzzy and multi-level problems in real life through mathematical language, and gives a scientific solution with a mathematical method. This article selects four operators, $M(\wedge, \vee)$, $M(\bullet, +)$, $M(\bullet, \vee)$ and $M(\wedge, \oplus)$, to calculate the standardized emergency decision matrix. The higher the comprehensive evaluation value, the better the solution.

The evaluation results of the scheme calculated based on the five mathematical models of the two models are shown in Table 22 and Figure 10. According to the results shown in the table, excluding the two failure operators $M(\wedge, \vee)$ and $M(\wedge, \oplus)$, the results obtained by the remaining TOPSIS method, $M(\bullet, +)$ operator and $M(\bullet, \vee)$ operator are $D_3 > D_2 > D_1$, while the result calculated by the prospect theory in this paper is $D_1 > D_3 > D_2$, which

is contradictory. Because decision-making information is one of the important bases for emergency decision-making in unconventional emergencies, the important content of emergency decision-making is the information collection, processing, feedback and other processes. Considering that the safety accident scenario of the project operation is not exactly the same, and different accidents have differences in weather, environment, human factors, etc., and the emergencies have the characteristics of a sudden incident, high time pressure and unprecedented reference, which determines that the decision-making information in the emergency situation is extremely lacking and difficult to obtain objective decision-making methods often cannot fully take into account the actual situation, and cannot achieve a correct and objective judgment based on the actual situation. Prospect theory considers subjective factors such as loss aversion of decision makers, and cognition of decision information among decision-makers varies. Therefore, in the face of highly uncertain information and incomplete information, the decision-making individuals in the decision-making group have different knowledge, experience, personality, etc., have different information cognition and risk cognition, different decision-makers' personal preferences and cognitive biases, so the final decision-making program selection has a certain impact. Therefore, based on the actual experience of the decision-maker, using prospect theory to select the program can get more convincing results.

**Table 22.** Comparison of calculation results between prospect theory and TOPSIS method and fuzzy comprehensive evaluation method.

| Scheme | Prospect Theory | TOPSIS | M (•, +) | M (∧, ∨) | M (•, ∨) | M (∧, ⊕) |
|--------|-----------------|--------|----------|----------|----------|----------|
| 1 | −0.1579 | 0.34163 | 0.31698 | 0.13500 | 0.04437 | 1.00000 |
| 2 | −0.5844 | 0.44087 | 0.32880 | 0.13500 | 0.04452 | 1.00000 |
| 3 | −0.2606 | 0.65511 | 0.35422 | 0.13500 | 0.05161 | 1.00000 |
| Result | $D_1 > D_3 > D_2$ | $D_3 > D_2 > D_1$ | $D_3 > D_2 > D_1$ | Invalidation | $D_3 > D_2 > D_1$ | Invalidation |

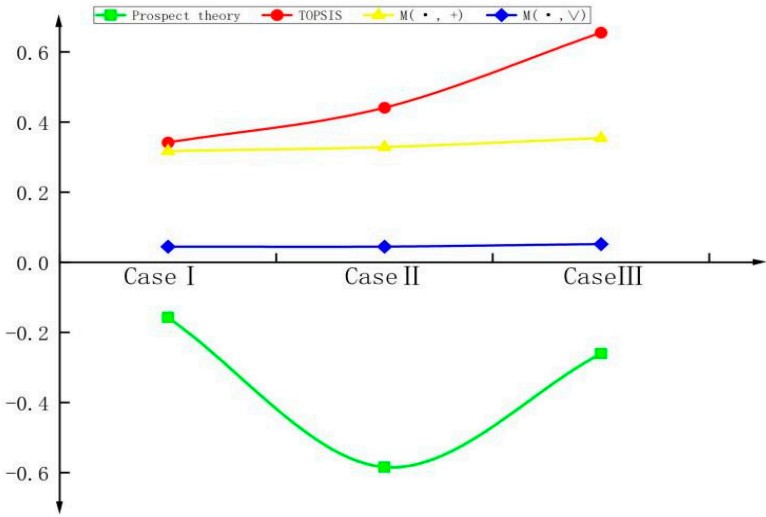

**Figure 10.** Comparison of the calculation results of prospect theory, the TOPSIS method and fuzzy comprehensive evaluation method.

In summary, the comparison results show that the proposed prospect theory method has a certain applicability and effectiveness for the decision-making of the engineering operation safety accident treatment scheme.

## 7. Conclusions

*7.1. Research Conclusions*

(1) Case representation is to classify the accident state characteristics of the basic content of the case, and express this by frame representation, XML representation, ontology representation and other methods. It is an effective way to describe the whole process of engineering accident handling.

(2) In the process of case reasoning, similar cases are initially found through case retrieval, then the similarity between cases is calculated through case attribute weights to obtain the primary selection scheme. This method is an effective way to screen out the similar scenes with the current accident from many accident scenes.

(3) In the prospect theory scheme optimization model based on the case reasoning conclusion, the decision information is normalized and the comprehensive prospect value of each scheme is calculated and the optimal scheme is obtained. This method is a decision-making method that considers subjective decision-making factors, which is more suitable for emergency decision-making scenarios, and is actually proved to be an effective method.

(4) This paper constructs an application case representation to digitize the accident scene and disposal measures; uses case reasoning structural similarity and overall similarity calculation to conduct preliminary screening, screen out accidents in similar scenarios, extract disposal plans from them; and finally, based on prospect theory, the final decision is made on the removed disposal plan, and the method system is suitable for the emergency decision-making problem of the South-to-North Water Diversion Project. At the same time, it also provides a specific reference value for similar emergency decision-making problems.

*7.2. Research Contributions and Future Research Directions*

7.2.1. Research Contributions

Based on the analysis of the characteristics of safety accidents in the middle route of the South-to-North Water Diversion Project, this paper constructs a preliminary selection model of the accident disposal scheme based on the case reasoning technology, and constructs an optimal model for the emergency scheme based on prospect theory and interval numbers.

The main contributions of this paper are:

1. A frame representation method of emergency cases of the water diversion project is constructed. The case is pre-classified according to the key attributes of the case by using the inductive index method, which reduces the search times of useful cases.

2. A two-layer search strategy based on the structural similarity and attribute similarity among cases is proposed, which overcomes the lack of attribute values in cases and effectively improves the search efficiency and accuracy.

3. An emergency plan optimization model based on prospect theory and interval number is constructed in this study, which replaces the utility function and probability in the expected utility theory with value function and weight function, making the calculation results reasonable and easy to understand, selecting the decision solution that best meets the psychological expectations of decision makers and improving the quality of emergency decision-making.

7.2.2. Future Research Directions

This paper uses frame notation to express the case, which brings about the loss of part of the information. In future research, we will consider using knowledge graphs for knowledge extraction and storage, and express them in a more precise semantic way to achieve more complex and refined reasoning.

**Author Contributions:** Conceptualization, F.L. and X.H.; methodology, X.H.; software, P.Z.; validation, P.Z., F.L. And X.H.; formal analysis, F.L.; investigation, Q.L.; resources, J.S.; data curation, J.S.; writing—original draft preparation, F.L.; writing—review and editing, X.H.; visualization, P.Z.; supervision, J.S.; project administration, Q.L.; funding acquisition, Q.L. All authors have read and agreed to the published version of the manuscript.

**Funding:** This research received no external funding.

**Institutional Review Board Statement:** Not applicable.

**Informed Consent Statement:** Not applicable.

**Data Availability Statement:** Not applicable.

**Conflicts of Interest:** The authors declare no conflict of interest.

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
