# Peer review of "Emergency Decision-Making for Middle Route of South-to-North Water Diversion Project Using Case-Based Reasoning and Prospect Theory"

_sustainability, doi:10.3390/su142113707_

Round 1

Reviewer 1 Report

The paper aims to improve the disposal efficiency of all kinds of accidents during the middle operation of the South-to-North Water Diversion Project, reduce people's property losses, and ensure the safety of water supply and put forward a new emergency decision-making method based on case-based reasoning technology and prospect theory. It adopts the prospect theory model to make the optimal decision for the accident disposal plan.

The content is succinctly described and contextualized with respect to previous studies. All references are correctly cited in the text and relevant to research. The authors are studied a significant number of studies and among them there are enough resentences from the last 5 years. 

The research process is well visualized through a figure, but I recommend authors to describe it in more details. The research method is appropriate and very well described.

Results are very well described and clearly presented in tables. To verify the differences between prospect theory and traditional objective methods authors apply two decision methods: TOPSIS method and fuzzy comprehensive evaluation method. The comparison of results confirms the applicability and effectiveness for the decision-making of the engineering operation safety accident treatment scheme.

The conclusions are supported by results. I recommend authors to state their plan for future work in the field in the conlusion.

Author Response

Response to comments

Thank you for your comments concerning our manuscript entitled “Research on Emergency Decision-Making Method of Middle Route of South-to-North Water Diversion Project Based on Case-Based Reasoning and Prospect Theory” (Manuscript ID: sustainability-1907385). Those comments are all valuable and very helpful for revising and improving our paper, as well as the important guiding significance to our researches. We have studied comments carefully and have made correction which we hope meet with approval. The main corrections in the paper and the responds to your comments are as follows:

1.The research process is well visualized through a figure, but I recommend authors to describe it in more details. The research method is appropriate and very well described.

Response:

This reviewer's suggestion made us realize that the article's explanation of the flow chart is not clear enough, so we made the following changes for 3.Research process:

This article uses a systematic process to illustrate the application process of the proposed method. We divide the method into three major pieces of content:

(1)Case representation. We use a specific method to represent all accident case scenarios, accident case disposal plans and accident case disposal effects as data. The case scenarios mainly include accident characteristic information, meteorological conditions, accident point information and surrounding environmental conditions; disposal plans include information such as rescue equipment, rescue teams, rescue materials, and specific measures; accident case disposal effects include accident disposal results, on-site Recovery status and lessons learned;

(2)Case-based reasoning scheme selection model. First, the historical accident cases are input into the case database, the first-level retrieval is carried out. Then historical case set is mainly screened according to the characteristic attributes of high discrimination; the overall similarity between the selected cases and the target case is compared. It consists of three parts: attribute similarity, structural similarity and attributes weight. Finally, several cases sets similar to the target accident are obtained;

(3)Scheme optimization model based on prospect theory. For the candidate case set selected by the primary selection model, standardize the data, set the reference point and the probability of the situation state, determine the probability weight function value, finally calculate the comprehensive prospect value of each scheme, and sort them to obtain the optimal scheme. The research process is shown in Figure 3.

2.The conclusions are supported by results. I recommend authors to state their plan for future work in the field in the conlusion.

Response:

Reviewer have given good suggestions for the conclusion of the article, and we revise and supplement the conclusion based on this suggestion to make the structure of the article more complete. Modify for 7.Conclusion:

  1. Conclusion

7.1 Research conclusions

(1)Case representation is to classify the accident state characteristics of the basic content of the case, and express it by frame representation, XML representation, ontology representation and other methods. It is an effective way to describe the whole process of engineering accident handling.

(2)In the process of case reasoning, similar cases are initially found through case retrieval, and then the similarity between cases is calculated through case attribute weights to obtain the primary selection scheme. This method is an effective way to screen out the similar scenes with the current accident from many accident scenes.

(3)Prospect theory scheme optimization model On the basis of the case reasoning conclusion, the decision information is normalized and the comprehensive prospect value of each scheme is calculated and the optimal scheme is obtained. This method is a decision-making method that considers subjective decision-making factors, which is more suitable for emergency decision-making scenarios, and is actually proved to be an effective method.

(4) This paper constructs an application case representation to digitize the accident scene and disposal measures, uses case reasoning structural similarity and overall similarity calculation to conduct preliminary screening, screen out accidents in similar scenarios, extract disposal plans from them, and finally based on prospect theory The final decision is made on the extracted disposal plan, and the method system is suitable for the emergency decision-making problem of the South-to-North Water Diversion Project. At the same time, it also provides a certain reference value for similar emergency decision-making problems.

7.2 Research contributions and future research directions

7.2.1 Research Contributions

Based on the analysis of the characteristics of safety accidents in the middle route of the South-to-North Water Diversion Project, this paper constructs a preliminary selection model of the accident disposal scheme based on the case reasoning technology, and constructs an optimal model for the emergency scheme based on prospect theory and interval numbers, and combines the evolution of different accident scenarios. The probability weights of each state are calculated, and the comprehensive prospect value of each scheme is finally obtained. The model provides an effective decision-making method for emergency safety decision-making.

7.2.2 Future research directions

This paper uses frame notation to express the case, which brings about the loss of part of the information. In the future research, we will consider using knowledge graphs for knowledge extraction and storage, and express them in a more precise semantic way to achieve more complex and refined reasoning.

Author Response

Response to comments

Thank you for your comments concerning our manuscript entitled “Research on Emergency Decision-Making Method of Middle Route of South-to-North Water Diversion Project Based on Case-Based Reasoning and Prospect Theory” (Manuscript ID: sustainability-1907385). Those comments are all valuable and very helpful for revising and improving our paper, as well as the important guiding significance to our researches. We have studied comments carefully and have made correction which we hope meet with approval. The main corrections in the paper and the responds to your comments are as follows:

  1. English is weak.

Response:

After clear opinions from reviewer, we read through the whole text and found some grammar and word errors and made corrections.

  1. References unrelated to the title are deleted.

Response:

Some literature citations in this paper are not closely related to the subject of the article. Therefore, in response to this problem, we have deleted the references that are not strongly related, and referenced the literature that is more in line with the subject of the article in the method introduction.

  1. I can't find a connection between table 6, then the theoretical presentation, then table 6.

Response:

Response:

The problems pointed out by Reviewer are not reflected in the fuzzy judgment matrix in detail in the article. The specific process is to invite experts to score the two index levels respectively, and then perform secondary processing according to the fuzzy scale in Table 4 to obtain the corresponding index level. The article 4.2.2. Calculation of the case attribute weight is modified as follows:

Suppose the index set of a certain layer is,then invite experts to score the two index levels respectively, and then perform secondary processing according to the fuzzy scale in Table 4, the fuzzy judgment matrix R of index set A can be obtained as:

  1. The fuzzy analytic hierarchy process is used to calculate the weight of each criterion: According to the importance of each index, the emergency decision-making management experts compared four indicators: economic cost, disaster site control degree, rescue timeliness and social impact, and obtained. How is it related to the case under discussion?The result is not clear, it is a fiction.

Response: 

The middle route of the South-to-North Water Diversion Project was opened in December 2014. Up to now, the specific situation of emergency safety incidents has been collected and analyzed. Since the quality and quantity of cases in the case database directly affect the validity of the cases given by the system through case retrieval, the increase of the number of cases enriches the content of cases. Continuous improvement of the basic information of the case can ensure that the research in this paper has more practical significance. This paper uses the case reasoning technology to construct a primary selection model of emergency plans. Using the case reasoning technology, the historical case set similar to the current accident can be quickly retrieved from the case database. The disposal plan saved in the historical case set can be used for the disposal of the current accident for reference. In view of the differences between similar cases and the current accident, the use of their disposal plans also has certain limitations. Therefore, on-site decision-makers need to revise the disposal plans of historical cases according to the accident conditions and make optimal decisions. Due to the existence of objective factors, such as the uncertainty of accident development and the incompleteness of decision-making information, as well as the profit preference and loss aversion psychology of decision-making experts, the decision-making of accident disposal will all have an impact, and prospect theory can effectively evaluate the subjective preference of decision-makers. Therefore, in this paper, the prospect theory model is used to optimize the accident disposal plan, and the optimal result can be expanded into the case database as a new disposal measure.

  1. In all the work, there is no application for your case. everything is exhaustive and useless. The conclusions are weak and do not show the connection between the cover and the content.

Response:

The emergency decision-making system developed with the algorithm of this study as the core is a part of China's key research and development plan topics (Topic 5 Topic 3: Research on Operation Safety Risk Warning and Disposal Measures of the South-to-North Water Diversion Project), and various functions have passed the third-party software test. It has been applied and achieved good results in the Huixian section of the middle route of the South-to-North Water Diversion Project. The application demonstration work adopts the form of field deployment and actual use by users to verify the demonstration content. The user side evaluates the functionality and applicability of the system. Because this paper mainly discusses the emergency decision-making algorithm based on CBR and prospect theory. Due to space limitations, the latter part of system integration is not included in this article. In the following research, it is hoped that the combination of knowledge graph and case reasoning will make the reasoning logic clearer.
